# Bayesian Active Learning for Bivariate Causal Discovery

**Yuxuan Wang** [1]  **Mingzhou Liu** [1]  **Xinwei Sun** [2]  **Wei Wang** [3]  **Yizhou Wang** [4]

## Abstract

Determining the direction of relationships between variables is fundamental for understanding complex systems across scientific domains. While observational data can uncover relationships between variables, it cannot distinguish between cause and effect without experimental interventions. To effectively uncover causality, previous works have proposed intervention strategies that sequentially optimize the intervention values. However, most of these approaches primarily maximized information-theoretic gains that may not effectively measure the reliability of direction determination. In this paper, we formulate the causal direction identification as a hypothesis-testing problem, and propose a Bayes factor-based intervention strategy, which can quantify the evidence strength of one hypothesis (*e.g.*, causal) over the other (*e.g.*, non-causal). To balance the immediate and future gains of testing strength, we propose a sequential intervention objective over intervention values in multiple steps. By analyzing the objective function, we develop a dynamic programming algorithm that reduces the complexity from non-polynomial to polynomial. Experimental results on bivariate systems, tree-structured graphs, and an embodied AI environment demonstrate the effectiveness of our framework in direction determination and its extensibility to both multivariate settings and real-world applications.

[1] School of Computer Science, Peking University, Beijing, China [2] School of Data Science, Fudan University, Shanghai, China [3] State Key Laboratory of General Artificial Intelligence, BIGAI, Beijing, China [4] School of Computer Science, Inst. for Artificial Intelligence, State Key Laboratory of General Artificial Intelligence, Peking University, Beijing, China. Correspondence to: Xinwei Sun <sunxinwei@fudan.edu.cn>, Wei Wang <wangwei@bigai.ai>.

*Proceedings of the 42nd International Conference on Machine Learning*, Vancouver, Canada. PMLR 267, 2025. Copyright 2025 by the author(s).

## 1. Introduction

Determining the direction of causal relationships between pairs of variables is a fundamental challenge in understanding complex systems. In fields like genetic research (Pinna et al., 2010; Jia & Zhang, 2022), while we may observe strong associations between genetic variants and phenotypes, determining which variable causally influences the other remains difficult. This directional uncertainty complicates our understanding of underlying mechanisms and limits our ability to make reliable predictions about system behavior.

Traditional methods for inferring causal direction rely primarily on observational data, representing relationships through directed edges between variable pairs. However, observational data alone often proves insufficient for determining causal direction due to the fundamental limitation of Markov equivalence (Pearl, 2009) - different causal directions can generate identical observational distributions, making them indistinguishable without intervention data.

The limitations of observational studies necessitate experimental interventions as the gold standard for determining causal direction (Fisher, 1936). While interventional experiments like gene knockouts have become increasingly feasible (Pinna et al., 2010), they remain costly and resource-intensive. This limitation has motivated the development of active learning methods for causal direction determination. Traditional approaches typically employ information-theoretic objectives (Tong & Koller, 2001; Murphy, 2001; Agrawal et al., 2019), greedily maximizing the mutual information between interventions and outcomes. However, recent studies (Gong et al., 2023) have shown that successful causal direction inference depends not only on the informativeness of generated data but also on its evidential complexity. This highlights a potential misalignment: maximizing information gain alone may not be optimal for bivariate causal discovery, as it does not directly optimize for the probability of drawing correct directional conclusions. This is a crucial distinction from a Bayesian decision-theoretic perspective (Trimmer et al., 2011), where the objective is typically to select actions that maximize the expected utility of the outcome, in our case, the correct identification of causal direction. Simply acquiring information is often a surrogate for, but not identical to, achieving a specific

decision-making goal.

In this paper, we study sequential intervention strategies to effectively determine causal direction while minimizing experimental costs. Working with a fixed intervention budget requires to consider intervention decisions jointly rather than independently (Huan & Marzouk, 2016; Lim et al., 2022), as myopic approaches focusing solely on immediate information gain may lead to suboptimal outcomes once the budget is exhausted. Our framework balances two key objectives: maximizing the probability of discovering the correct causal direction early in the sequence, while maintaining the capability to reach definitive conclusions by the end of the budget period. This requires carefully planning each intervention by considering both its immediate evidential value and its impact on future discovery potential. Moreover, sequential optimization is computationally expensive, especially when the intervention budget or the action space is large, necessitating efficient algorithmic solutions.

To address these challenges, we propose a framework that employs Bayes factors (Kass & Raftery, 1995) to formulate an objective for edge orientation. This approach is inherently decision-focused: Bayes factors enable systematic hypothesis testing by quantifying the relative evidence supporting competing directional hypotheses, directly informing the decision about causal direction. This formulation allows us to explicitly optimize for obtaining decisive and correct directional evidence—aligning with established thresholds for when evidence is deemed conclusive—rather than pursuing broader informational criteria that may not be as efficiently targeted towards the specific decision at hand. We develop a sequential objective that explicitly balances immediate and future rewards, and use dynamic programming to efficiently optimize the intervention selection process, ensuring that each intervention not only provides immediate insights but also maintains flexibility for future direction determinations. While the framework is introduced for bivariate cases, it extends to tree-structured graphs (Greenewald et al., 2019), enabling root cause localization through sequential interventions.

To demonstrate the effectiveness and generality of our framework, we evaluate our methods on bivariate edge orientation tasks and further extend to more challenging scenarios including root source localization in tree-structured graphs (Greenewald et al., 2019) and a switch-light reasoning task in embodied AI environments (Peng et al., 2024). Our experiments on tree-structured graphs demonstrate the potential for handling multivariate settings through sequential interventions, particularly in applications such as biology (Jia & Zhang, 2022) and system anomaly detection (Han et al., 2023). Additionally, results from the embodied AI experiment show that our method can efficiently make correct decisions about causal relationships in physical environ-

ments. These results validate the framework potential for both scaling to complex multivariate systems and practical deployment in real-world scenarios.

The main contributions of this paper are:

1. We propose an active intervention framework that leverages Bayes factors to quantify the strength of directional evidence, providing an alternative to traditional information-theoretic approaches.

2. We formulate a sequential objective that balances short-term and long-term intervention outcomes, and develop a dynamic programming algorithm for efficient optimization of intervention selection.

3. We evaluate our method on bivariate edge orientation tasks, root source localization in tree-structured graphs (Greenewald et al., 2019), and a switch-light reasoning task in embodied AI settings.

## 2. Related work

**Active causal discovery.** Traditional causal discovery methods based on observational data face the inherent challenge of the Markov equivalence class (Pearl, 2009), where multiple causal structures can induce identical observations. To overcome this constraint, researchers have developed active intervention techniques. For example, (He & Geng, 2008) proposed to maximize edge recovery per intervention; (Shanmugam et al., 2015) selected the optimal intervention vertices based on graph coloring. Works that are most relevant to ours are (Murphy, 2001; Tong & Koller, 2001; Masegosa & Moral, 2013; Agrawal et al., 2019; Toth et al., 2022), which selected interventions that maximize the mutual information between the graph and hypothetical samples. However, such information-theoretic objectives can be suboptimal from a Bayesian decision-theoretic standpoint (Trimmer et al., 2011) when the goal is a specific, correct determination, as they do not always maximize the probability of achieving confident causal conclusions (Greenewald et al., 2019). **In contrast** to purely information-theoretic goals, we employ Bayes factors to directly optimize for decisive and accurate evidence for a specific causal hypothesis, aiming for more efficient discovery for this targeted decision.

**Bivariate causal discovery.** To identify the causal direction between two variables, previous works leverage asymmetry prior between the cause and the effect for identification through only observational data. Upon this prior, one line of works employed Kolmogorov complexity as a measure of causal direction (Tagasovska et al., 2020; Mitrovic et al., 2018). However, the Kolmogorov complexity is computationally intractable. Another line of work imposed restrictions on the functional class of causal relationships, such

as assuming additive noise models (Hoyer et al., 2008; von Kügelgen et al., 2019; Blöbaum et al., 2018). However, it may fail to generalize to a broader family of models. **In this paper**, we achieve bivariate causal identification through active intervention, without imposing any asymmetry constraints to the causal relationship.

**Sequential experiment design (SED).** SED proposes to generate the design points in a sequential manner until the convergence of optimization, in order to overcome the oversampling or undersampling issue met in design of experiments (DOE) (Rainforth et al., 2024). For example, (Chakraborty & Chowdhury, 2016) introduced a novel distribution adaptive sequential experimental design for generalised analysis of variance; (Foster et al., 2021) reduced the computational cost of traditional Bayesian DOE by learning an amortized deep design network; and (Blau et al., 2022) induced the design policy by solving a Markov decision process. Particularly, in causal discovery, researchers such as (Tigas et al., 2022; Toth et al., 2022) proposed to apply SED using information-theoretic criteria, aiming to maximize information gain with each intervention. However, information gain maximization can be suboptimal for achieving specific, correct decisions. For decision-focused SED, Bayesian decision theory (Reggiani & Weerts, 2008; Berger, 2013) offers an alternative by optimizing choices for desired outcomes or minimal loss. This aligns with broader needs in tasks like sequential decision-making under causal uncertainty (Gonzalez-Soto et al., 2018), where robust causal models guide subsequent actions. **In this paper** aligning with a decision-theoretic approach, we introduce an SED that explicitly gathers information for obtaining decisive and correct evidence regarding causal direction, making the optimization more effective for achieving correct causal conclusions.

## 3. Preliminaries

**Problem setup & Notations.** Consider a bivariate system with $X \in \mathcal{X}$ and $Y \in \mathcal{Y}$, our goal is to design an intervention strategy to effectively distinguish the causal relation $\mathbb{H}_1 : X \to Y$ from the anti-causal one $\mathbb{H}_0 : X \leftarrow Y$. To achieve this, we assume the availability of $n$ observational data $\mathcal{D}_{\text{obs}} = \{(x_1, y_1), \cdots, (x_n, y_n)\}$ that are *i.i.d* drawn from the joint distribution $\mathbb{P}(X, Y)$. When $X \to Y$, the underlying structural causal model (SCM) is:

$$x_i := f_X(N_{X,i}), \quad y_i := f_Y(x_i, N_{Y,i}), \; \forall i,$$

where $f_X$ and $f_Y$ are structural equations, and $\{N_{X,i}\}_{i=1}^n$, $\{N_{Y,i}\}_{i=1}^n$ are independent noise. Throughout, we assume $N_X$ and $N_Y$ are d-separated in the graph, *i.e.*, $N_X \perp\!\!\!\perp_d N_Y$, which implies the causal sufficiency condition (Spirtes et al., 2001) that is standard in causal discovery. For notational convenience, we denote $\mathbb{N}$ the set of natural numbers, and

denote $1 : k = \{1, 2, \ldots, k\}$ for any $1 \le k \in \mathbb{N}$. For a vector $\boldsymbol{x} \in \mathbb{R}^K$, we denote $\boldsymbol{x}_{-k}$ as the sub-vector that excludes the $k$-th element.

**Bayes factor.** To measure the uncertainty during the intervention, we borrow the concept of *Bayes factor* (Jeffreys, 1961; Kass & Raftery, 1995) defined as the likelihood ratio $\text{BF}_{01} = \mathbb{P}(\mathcal{D}|\mathbb{H}_0)/\mathbb{P}(\mathcal{D}|\mathbb{H}_1)$ over data $\mathcal{D}$, which measures the strength of evidence in favor of the null hypothesis over the alternative. It has been widely used in selecting the true graph among Markov equivalence classes (Madigan et al., 1996; Castelo & Perlman, 2004; Castelletti & Consonni, 2024). In particular, (Castelletti & Consonni, 2024) employed the Bayes factor in active causal discovery; however, it only used it to determine the minimum number of interventions, without truly interacting with the environment. **In this paper**, we provide a framework that incorporates the Bayes factor to actively interact with the environment, enabling more effective identification of the causal relationship.

## 4. Methodology

In this section, we present our intervention strategy for learning causal direction. In section 4.1, we first introduce the Bayesian objective over a single-step intervention for interaction. To interact, section 4.2 introduces an iterative algorithm, including prior initialization, optimization, and posterior updating. Section 4.3 extends the framework to multi-step interventions. Finally, section 4.4 applies our approach to multivariate causal discovery.

### 4.1. Bayesian sample selection

Suppose we have conducted $m$ interventions so far, and the corresponding intervention data are denoted as $\mathcal{D}_m := \{(\overline{x}_1, \overline{y}_1), \ldots, (\overline{x}_{n_0}, \overline{y}_m)\}$. To determine the next intervention value $\overline{x}$, we propose to maximize the probability of *decisive* and *correct* evidence (denoted as $\mathbb{P}_{\text{DC}}$):

$$\mathbb{P}_{\text{DC}}(\overline{x}) = \mathbb{P}_{\text{DC}}^0(\overline{x}) \cdot \mathbb{P}(\mathbb{H}_0) + \mathbb{P}_{\text{DC}}^1(\overline{x}) \cdot \mathbb{P}(\mathbb{H}_1) \quad \text{(1a)}$$

$$\mathbb{P}_{\text{DC}}^0(\overline{x}) = \mathbb{P}\left\{\text{BF}_{01}\left(\mathcal{D}_m \cup \{\overline{x}, \overline{y}\}\right) > k_0 | \mathbb{H}_0, \mathcal{D}_m\right\} \quad \text{(1b)}$$

$$\mathbb{P}_{\text{DC}}^1(\overline{x}) = \mathbb{P}\left\{\text{BF}_{01}\left(\mathcal{D}_m \cup \{\overline{x}, \overline{y}\}\right) < k_1 | \mathbb{H}_1, \mathcal{D}_m\right\}. \quad \text{(1c)}$$

The objective $\mathbb{P}_{\text{DC}}$ is the weighted sum of the probability of determination under the null and alternative hypothesis. The Bayes factor is defined as:

$$\text{BF}_{01}(\mathcal{D}_m) = \frac{\mathbb{P}(\mathcal{D}_m \mid \mathbb{H}_0)}{\mathbb{P}(\mathcal{D}_m \mid \mathbb{H}_1)}.$$

Therefore, the objective (1a) takes expectation over $p(\overline{y})$ under $\mathbb{H}_0$, and $p\{\overline{y}|\text{do}(\overline{x})\}$ under $\mathbb{H}_1$. Since we cannot identify causal directions through only $\mathcal{D}_{\text{obs}}$, we set the prior $\mathbb{P}(\mathbb{H}_0) = \mathbb{P}(\mathbb{H}_1) = 1/2$. Here, $k_0, k_1$ are related to the decisive evidence under $\mathbb{H}_0$ and $\mathbb{H}_1$, respectively.

Specifically, we say the data provide decisive evidence in favor of $\mathbb{H}_0$ at level $\gamma_0$ if $\mathrm{BF}_{01}(\mathcal{D}) > k_0 := \frac{\gamma_0}{\omega(1-\gamma_0)}$ with $\omega := \frac{\mathbb{P}(\mathbb{H}_0)}{\mathbb{P}(\mathbb{H}_1)}$, since we have $\mathbb{P}(\mathbb{H}_0|\mathcal{D}) > \gamma_0$.

The objective has been similarly used to determine the sample size for interventions (Castelletti & Consonni, 2024), without actually interacting with the environment. In contrast, we consider the scenario where we can interact with the environment. During the interaction, our goal is sequentially choosing the values of $X$ for intervention to most effectively reduce the uncertainty about the causal direction.

**Connection to information gain.** One may consider optimizing the intervention sample by maximizing the information gain (Lindley, 1956; Bernardo, 1979) that has been employed in causal discovery (Mian et al., 2023; Toth et al., 2022). We have demonstrated in Appx. A that the information gain is a monotonic transformation of the Bayes factor, meaning it may favor interventions that maximize the Bayes factor given each hypothesis. However, the Bayes factor is more related to the causal hypothesis testing considered in our scenario, it is therefore employed to guide the intervention process.

### 4.2. Optimization and posterior update

We introduce how to optimize $\overline{x}$ and update the causal belief at each iteration. Our procedure involves *estimating the initial intervention distribution* for calculating the Bayes factor, *optimization of $\overline{x}$*, and *posterior update*.

**Estimating the intervention distribution.** To compute the Bayes factor in (1a), we need to compute $p\{y|\mathrm{do}(x), \mathbb{H}_j\}$ under each hypothesis, which equals to:

$$p\{y|\mathrm{do}(x), \mathbb{H}_0\} = \int p(y \mid \theta_0)p(\theta_0|\mathbb{H}_0)d\theta_0; \quad (2)$$

$$p\{y|\mathrm{do}(x), \mathbb{H}_1\} = \int p(y|x, \theta_1)p(\theta_1|\mathbb{H}_1)d\theta_1, \quad (3)$$

since the edge $Y \to X$ is broken when we intervene on $x$, we have $p\{y|do(x)\} = p(y)$ under $\mathbb{H}_0$. Under $\mathbb{H}_1$, we have $p\{y|do(x)\} = p(y|x)$. For each hypothesis, we take the expectation over $p(\theta_j|\mathbb{H}_j)$. Following (Kass & Raftery, 1995), we specify $p(\theta_j|\mathbb{H}_j)$ as the data-dependent prior centered around the MLE estimate. when $X, Y$ are continuous, we set it as Gaussian distribution $\mathcal{N}(\widehat{\theta}, \widehat{\Sigma})$, where $\widehat{\theta}$ is the MLE estimate, and $\widehat{\Sigma}$ is the inverse Hessian of the negative log-likelihood. When $X, Y$ are categorical, we use Beta distribution to specify each level's parameter, where the parameters in Beta distribution is obtained as the MLE estimate. Suppose $X, Y$ respectively have $l_X$ and $l_Y$ levels, the parameters we specify include $\theta_{0,i} := \mathbb{P}(Y = i)$ for $i < l_X$ under $\mathbb{H}_0$, and $\theta_{1,j,i} := \mathbb{P}(Y = j|X = i)$ for $j < l_Y$ and $i \le l_X$ under $\mathbb{H}_1$. For each parameter $\theta_{0,i} := \mathrm{Beta}(\alpha_{0,i}, \beta_{0,i})$ (the prior of $\theta_{1,j,i}$ is similar), where $\alpha_{0,i}, \beta_{0,i}$ are MLE estimates

from observational data.

In this paper, we do not update these priors using interventional data. This is because, even when $\mathbb{H}_0$ holds, the data may be better fit by $p(\theta_0|\mathbb{H}_1)$, making it difficult for the Bayes factor to provide sufficient evidence to support $\mathbb{H}_0$. While we can put constraints on $\theta_1$ to avoid this issue, it is not necessary in the scenario considered in this paper.

**Optimization.** Having identified the interventional distributions, we now focus on optimizing $\mathbb{P}_{\mathrm{DC}}$ over $\overline{x}$. To this end, we expand $\mathbb{P}_{\mathrm{DC}}^0$ and $\mathbb{P}_{\mathrm{DC}}^1$ in (1) with the following form:

$$\mathbb{P}_{\mathrm{DC}}^0(\overline{x}) = \int_{\mathcal{Y}} \mathbb{I}\{\mathrm{BF}_{01}(\mathcal{D}_m \cup \{(\overline{x}, \overline{y})\}) > k_0\}$$
$$\times \, p\{\overline{y}|\mathrm{do}(\overline{x}), \mathbb{H}_0\}d\overline{y}\mathbb{P}(\mathbb{H}_0|\mathcal{D}_m); \quad (4)$$

$$\mathbb{P}_{\mathrm{DC}}^1(\overline{x}) = \int_{\mathcal{Y}} \mathbb{I}\{\mathrm{BF}_{01}(\mathcal{D}_m \cup \{(\overline{x}, \overline{y})\}) < k_1\}$$
$$\times \, p\{\overline{y}|\mathrm{do}(\overline{x}), \mathbb{H}_0\}d\overline{y}\mathbb{P}(\mathbb{H}_1|\mathcal{D}_m). \quad (5)$$

This objective integrates over all possible outcomes $\overline{y}$ after intervention. When $X, Y$ are continuous, we employ Monte-Carlo approximation to estimate the objective and apply a smooth approximation of the indication function, which enables gradient-based optimization. When $X, Y$ are discrete, we can directly evaluate the objective by enumerating all possible intervention values to find the optimum. Please refer Appendix B.3 for details.

**Posterior update.** Having obtained the optimal $\overline{x}$, we get $\overline{y}$ through interaction. We can append the interventional data $\mathcal{D}_{m+1} = \mathcal{D}_m \cup \{\overline{x}, \overline{y}\}$. With these data, we update our causal belief, *a.k.a*, posterior of each hypothesis as follows:

$$\mathbb{P}(\mathbb{H}_j|\mathcal{D}_{m+1}) = \frac{p(\{\overline{x}, \overline{y}\}|\mathbb{H}_j)\mathbb{P}(\mathbb{H}_j|\mathcal{D}_m)}{\sum_{j \in \{0,1\}} p(\{\overline{x}, \overline{y}\}|\mathbb{H}_j)\mathbb{P}(\mathbb{H}_j|\mathcal{D}_m)}$$
$$= \frac{p(\overline{x})p(\overline{y}|\overline{x}, \mathbb{H}_j)\mathbb{P}(\mathbb{H}_j|\mathcal{D}_m)}{\sum_{j \in \{0,1\}} p(\overline{x})p(\overline{y}|\overline{x}, \mathbb{H}_j)\mathbb{P}(\mathbb{H}_j|\mathcal{D}_m)},$$

where $p(\overline{y}|\overline{x}, \mathbb{H}_j)$ are given by (2-3) for $j = 0, 1$.

We summarize the whole procedure in Alg. 1 with $k = 1$ in lines 6-7.

### 4.3. Multi-step optimization

While the single-step intervention design presented in section 4.2 provides a foundation, its greedy and myopic nature may lead to suboptimal solutions (Rainforth et al., 2024). In this section, we propose a multi-step version that optimizes a sequence of interventions by considering cumulative evidence across multiple steps, along with an efficient algorithm for solving the resulting optimization problem.

**Multi-step objective.** Following (González et al., 2016; Jiang et al., 2020), we optimize the next step by also looking

---

**Algorithm 1** Bayesian active intervention

1: **Input:** Observational data $\mathcal{D}_{\text{obs}}$, total budget $B$
2: **Output:** $\mathcal{D}_{\text{int}}$, posterior $\mathbb{P}(\mathbb{H}_j|\mathcal{D}_{\text{int}})$ for $j = 0, 1$.
3: Initialize $\mathcal{D}_{\text{int}} = \emptyset$, $\mathbb{P}(\mathbb{H}_0) = \mathbb{P}(\mathbb{H}_1) = \frac{1}{2}$.
4: Specify $\mathbb{P}(\theta|\mathbb{H}_j)$ from $\mathcal{D}_{\text{obs}}$ for $j \in \{0, 1\}$.
5: **while** $|\mathcal{D}_{\text{int}}| < B$ **do**
6:   Set remaining budget $K = B - |\mathcal{D}_{\text{int}}|$.
7:   Obtain $\{x_t\}_{t=1}^{K}$ by maximizing $\sum_{i=1}^{K} \gamma^k \mathbb{P}_{\text{DC}}(\overline{x}_{1:i})$.
8:   Perform $\text{do}(X = \overline{x}_1)$ and observe $\overline{y}$.
9:   Update $\mathcal{D}_{\text{int}} \leftarrow \mathcal{D}_{\text{int}} \cup \{(x_1, y)\}$.
10:   Update posterior $\mathbb{P}(\mathbb{H}_j|\mathcal{D}_{\text{int}})$.
11: **end while**

---

ahead its next $K - 1$ steps (for simplicity, we omit $\mathcal{D}_m$):

$$\mathbb{P}_{\text{DC}}(\overline{x}_{1:K}) = \mathbb{P}\{\text{BF}_{01}(\{\overline{x}_1, \overline{y}_1\} \cup \mathcal{D}_{\text{fut}}) > k_0|\mathbb{H}_0\}\mathbb{P}(\mathbb{H}_0)$$
$$+ \mathbb{P}\{\text{BF}_{01}(\{\overline{x}_1, \overline{y}_1\} \cup \mathcal{D}_{\text{fut}}) < k_1|\mathbb{H}_1\}\mathbb{P}(\mathbb{H}_1), \quad (6)$$

where $\mathcal{D}_{\text{fut}} = \{\overline{z}_2, ..., \overline{z}_K\}$ represents data of the next $K - 1$ steps, with $\overline{z}_i := (\overline{x}_i, \overline{y}_i)$. To determine the optimal $\overline{x}_1$ for intervention, we need to optimize over $\overline{x}_{1:K}$. Here, $K$ is a trade-off between the short-term and long-term benefits. Although a large value of $K$ would maximize certainty in the long run, it may affect the effectiveness in the immediate steps. To balance the short-term and long-term benefits, we propose to optimize the following objective:

$$J(\overline{x}_1, \cdots, \overline{x}_K) := \sum_{k=1}^{K} \gamma^k \mathbb{P}_{\text{DC}}(x_{1:k}), \quad (7)$$

where $0 < \gamma < 1$ denotes the discount factor. Such a design is common in sequential experimental design (Foster et al., 2021) and reinforcement learning (Huan & Marzouk, 2016) that balances immediate and future rewards. However, our work differs in its optimization target: instead of maximizing information gain or expected utility, we directly optimize the probability of reaching decisive causal conclusions. This arises from the practical goal that determines causal relationships within limited experimental budgets.

**Optimization via dynamic programming.** We consider both categorical and continuous cases for optimizing (7).

When $X$ is continuous, we can follow Sec. 4.2 to employ Monte Carlo approximation for the smooth approximation of the objective. When $X$ is discrete, evaluating (7) requires considering $|\mathcal{X}|^k$ possible sequences, which is computationally infeasible when $k$ is large. To accelerate, we use dynamic programming, which builds upon two key properties that allow for efficient optimization, as presented below.

**Proposition 4.1.** *Suppose $X$ has $l_X$ categories, $\widetilde{x}_1, ..., \widetilde{x}_{l_X}$. For any $k$, $\mathbb{P}_{\text{DC}}(\overline{x}_{1:k})$ is symmetric with respect to its arguments $(\overline{x}_1, \cdots, \overline{x}_k)$.*

This property implies that $\mathbb{P}_{\text{DC}}(\overline{x}_{1:k})$ depends only on the count of each value from $\mathcal{X}$ that appears in the sequence $\overline{x}_{1:k}$. Therefore, we can rewrite $J(\overline{x}_1, ..., \overline{x}_K)$ as $J(\boldsymbol{n}_K)$ and $\mathbb{P}_{\text{DC}}(\overline{x}_{1:k})$ as $\mathbb{P}_{\text{DC}}(\boldsymbol{n}_k)$, where $\boldsymbol{n}_k \in |\mathbb{N}|^{l_X}$ for any $k \leq K$ represents the count vector. The $i$-th element of $\boldsymbol{n}_k$ records how many times the value $\widetilde{x}_i$ appears in the sequence $\overline{x}_{1:k}$. Upon this property, the following proposition ensures us to optimize $J(\overline{x}_{1:K})$ via dynamic programming.

**Proposition 4.2.** *Denote $\overline{x}_{1:K}^*$ as the optimal value of $J(\overline{x}_{1:K})$. Given $\overline{x}_{1:k-1}^*$ for each $k \leq K$, the optimal value of $\overline{x}_k^*$ only depends on $\boldsymbol{n}_{k-1}^*$.*

The proof is left to Appx. B.2. This property ensures us to solve $\overline{x}^*$ in a recursive manner:

$$J_k(\boldsymbol{n}_k) := \max_{x_k} \{\mathbb{P}_{\text{DC}}(\boldsymbol{n}_k) + \gamma J_{k-1}(\boldsymbol{n}_k - \boldsymbol{e}_{\overline{x}_k})\}$$
$$\overline{x}_k^* := \arg\max_{\overline{x}_k} \{\mathbb{P}_{\text{DC}}(\boldsymbol{n}_k) + \gamma J_{k-1}(\boldsymbol{n}_k - \boldsymbol{e}_{\overline{x}_k})\}$$
$$(8)$$

for each $k \leq K$, where $\boldsymbol{e}_{\overline{x}_k} \in \{0, 1\}^{l_X}$ represents the one-hot encoded vector, where the element corresponding to $\overline{x}_k$ is 1, and all other elements are 0. In this recursive formulation, the optimal solution to the original problem is composed of the optimal solution to each subproblem.

**Complexity Analysis.** For a given remaining intervention budget $K$, our algorithm determines the optimal intervention sequence $(x_1^*, ..., x_K^*)$ that maximizes the cumulative reward $J(\overline{x}_{1:K})$. Since both $x^*$ and $J$ are determined by count vectors $\boldsymbol{n}$, we first enumerate all possible count vectors and compute their corresponding $\mathbb{P}_{\text{DC}}$ values **(Phase 1)**, then recursively compute the optimal cumulative rewards $J_k(\boldsymbol{n})$ for each $k$ **(Phase 2)**. Finally, we obtain $\boldsymbol{x}^*$ through backward optimization **(Phase 3)**. The complexity analysis for each phase proceeds as follows.

**Phase 1: State Precomputation (lines 4-6).** This phase enumerates all feasible count vectors $\boldsymbol{n} = (n_1, ..., n_{|\mathcal{X}|})$ where $n_x \in \mathbb{N}$ denotes the number of applications for intervention $x \in \mathcal{X}$, constrained by $\sum_{x \in \mathcal{X}} n_x \leq K$. This represents all possible ways to distribute $K$ interventions across $|\mathcal{X}|$ different intervention types, with the total number of combinations being $\binom{K+|\mathcal{X}|-1}{|\mathcal{X}|-1} = O(K^{|\mathcal{X}|-1})$. Each state's $\mathbb{P}_{\text{DC}}$ value is precomputed and stored for subsequent optimization.

**Phase 2: Recursive State Optimization (lines 8-12).** For each step $k = 1, ..., K$, the algorithm computes optimal cumulative rewards $J_k(\boldsymbol{n})$ through backward induction: **i)** *State Generation*: Enumerate all count vectors with $\sum_x n_x = k$, requiring $O(k^{|\mathcal{X}|-1})$ states. **ii)** *State Update*: Evaluate all possible interventions $x \in \mathcal{X}$ using the recursive relation (Equation (8)) with $O(|\mathcal{X}|)$ operations per state. Aggregating across all budget levels yields: $\sum_{k=1}^{K} O(|\mathcal{X}| \cdot k^{|\mathcal{X}|-1}) = O(|\mathcal{X}| \cdot K^{|\mathcal{X}|})$.

**Phase 3: Optimal Path Reconstruction (lines 14-18).**
Starting from the terminal state $\boldsymbol{n}_K^*$, the optimal intervention sequence is reconstructed by iteratively subtracting unit vectors: $\boldsymbol{n}^* \leftarrow \boldsymbol{n}^* - \boldsymbol{e}_{x_t^*}, \quad k = K, \ldots, 1$ This backtracking process requires $O(K)$ operations, independent of $|\mathcal{X}|$.

**Total Complexity.** Combining all phases yields: $O(K^{|\mathcal{X}|}) + O(|\mathcal{X}| \cdot K^{|\mathcal{X}|}) + O(K) = O(|\mathcal{X}| \cdot K^{|\mathcal{X}|})$. which is a polynomial complexity for $K$. This represents an exponential improvement over the naive $O(|\mathcal{X}|^K)$ brute-force approach.

---

**Algorithm 2** Dynamic Programming for Optimal Intervention Design

---

1: **Input**: Remaining budget $K$, intervention space $\mathcal{X}$
2: **Output**: Optimal intervention sequence $(x_1^*, ..., x_K^*)$
3: *Phase 1: Precompute* $\mathbb{P}_{\mathrm{DC}}$ *values*
4: **for** all count vectors $\boldsymbol{n}$ with $\sum_{x \in \mathcal{X}} n_x \leq K$ **do**
5:     Store $\mathbb{P}_{\mathrm{DC}}(\boldsymbol{n})$ {Compute via causal model}
6: **end for**
7: *Phase 2: Forward value iteration*
8: **for** each budget $k = 1$ to $K$ **do**
9:     **for** all feasible $\boldsymbol{n}_k$ with $\sum_x n_x = k$ **do**
10:         Compute $J_k(\boldsymbol{n}_k)$ via Equation (8)
11:     **end for**
12: **end for**
13: *Phase 3: Backward path retrieval*
14: Initialize $\boldsymbol{n}^* \leftarrow \arg\max_{\boldsymbol{n}} J_K(\boldsymbol{n})$
15: **for** $k = K$ down to 1 **do**
16:     Find $x_k^*$ that maximized Equation (8) for $\boldsymbol{n}^*$
17:     Update $\boldsymbol{n}^* \leftarrow \boldsymbol{n}^* - \boldsymbol{e}_{x_k^*}$
18: **end for**

---

### 4.4. Application to multivariate causal discovery

Our approach can be naturally applied to causal discovery over multiple variables, when combined with off-the-shelf approaches in identifying the target variable for intervention (He & Geng, 2008; Hauser & Bühlmann, 2014). For such target variable, it may be associated with multiple causal directions to determine. We can therefore optimize the value by combining the objective of all directions of interest. Besides, since we do not update $p\{y|do(x)\}$ once it is fixed, we require it to be identifiable. To illustrate, we provide examples over three variables in Appx. C.

**Causal trees.** The type of causal graph that naturally satisfies this identifiable condition is causal trees, which can be widely applied in various domains, such as microservice systems diagnostics (Xin et al., 2023; Pham et al., 2024; Fariha et al., 2020), general system anomaly detection (Han et al., 2023; Palanki, 2024).

Specifically, given a tree skeleton with vertex set $\mathcal{V}$ and edge set $\mathcal{E}$, existing methods like (Greenewald et al., 2019) provide strategies for selecting intervention targets. For each

selected intervention target $V \in \mathcal{V}$, our goal is to determine which neighboring node is its parent, thereby identifying edge directions in the causal tree. Let $\mathcal{N}(V)$ denote the neighbors of node $V$ in the tree skeleton. We design optimal intervention values $\overline{x}_V$ by maximizing (1) that is associated with multiple hypothesis:

$$\sum_{U \in \mathcal{N}(V)} \mathbb{P}_{\mathrm{DC}}(\overline{x}_V; U),$$

where $\mathbb{P}_{\mathrm{DC}}(\overline{x}_V; U)$ denotes the objective whose goal is to discriminate $\mathbb{H}_1 : V \to U$. from its null $\mathbb{H}_0 : V \leftarrow U$.

## 5. Experiments

In this section, we evaluate our method on bivariate causal discovery and tree-structure causal graph learning (Greenewald et al., 2019). We also consider a switch-light reasoning task to demonstrate the usage of our method in embodied artificial intelligence (Embodied AI).

**Compared baselines.** We compare our method with the following baselines: **i) Rand** that randomly selects the intervention value; **ii) InfoGain** (Tigas et al., 2022; Toth et al., 2022) that optimizes the information gain to choose the best intervention value. To accommodate this method to our hypothesis testing setting, we compute the Bayes factor after optimization to inform the decision; and **iii) Multistep InfoGain** that enhances (Tigas et al., 2022; Toth et al., 2022) by incorporating the multi-step optimization scheme described in Sec. 4.3.

**Evaluation metrics.** For bivariate causal discovery, we report the rate of type I error and recall, which are the probabilities of incorrectly rejecting the null hypothesis when it is true, and correctly rejecting it when it is false, respectively. We also report the probability of decisive and correct evidence $\mathbb{P}_{\mathrm{DC}}$ defined in (1). For tree-structure causal discovery and the switch-light reasoning task, we report the total number of interventions required to recover the causal graph.

**Implementation details.** We set the evidence levels in (1b) and (1c) to $k_0 = 10$ and $k_1 = 1/10$, respectively. Results for alternative parameter settings, including $k_0 = 30, k_1 = 1/30$ and $k_0 = 100, k_1 = 1/100$, are presented in D.3. The selection of these parameter values is based on the classification of Bayes factor detailed in Appendix D.1, following (Kass & Raftery, 1995; Schönbrodt & Wagenmakers, 2018). For Alg. 1, we set the total budget to $B = 100$ and the number of observational samples to $|\mathcal{D}_{\mathrm{obs}}| = 1000$. For the optimization over discrete variables, we consider Alg. 2. For the optimization over continuous variables, we employ the Adam optimizer with a learning rate of 0.1 and a total iterations of 4000 steps. We decay the learning rate to 0.001 after the first 200 steps. The switch-light reasoning task

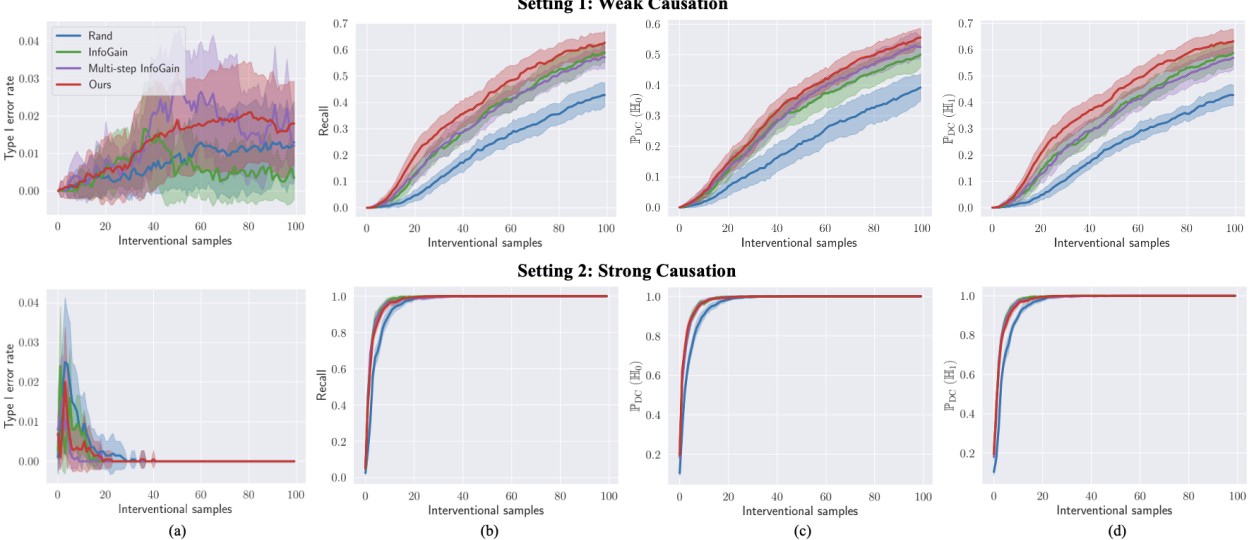

*Figure 1.* Results on bivariate causal discovery. The two rows presents results in two different settings. Columns (a)-(d) report the type I error rate, recall, $\mathbb{P}_{DC}$ under $\mathbb{H}_0$, and $\mathbb{P}_{DC}$ under $\mathbb{H}_1$, respectively, in relation to the number of intervention samples.

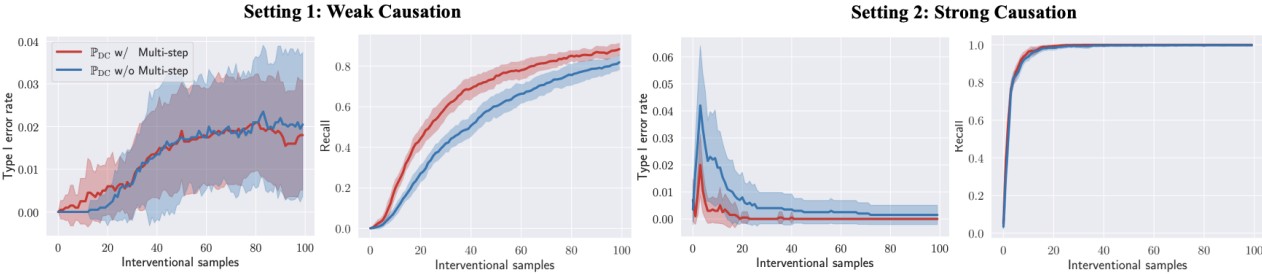

*Figure 2.* Ablation study on the multi-step optimization algorithm. The red and blue lines denote our method with and without multi-step optimization, respectively. The first two columns are results from Setting-1, and the last two columns are results from Setting-2.

is implemented on the TongSim (Peng et al., 2024) engine running on a server with NVIDIA 2080-Ti GPUs.

### 5.1. Bivariate causal discovery

**Data generation.** We consider binary variables[1], where the data is generated from $\mathbb{P}(Y) \sim \text{Bern}(0.5)$ (resp. $\mathbb{P}(X)$) and $\mathbb{P}(X|Y=0) \sim \text{Bern}(p_0)$, $\mathbb{P}(X|Y=1) \sim \text{Bern}(p_1)$ (resp. $\mathbb{P}(Y|X)$) under the null hypothesis $\mathbb{H}_0$ (resp. the alternative hypothesis $\mathbb{H}_1$). Here, the quantity $|p_0 - p_1| = \text{TV}(\mathbb{P}(X|Y=0), \mathbb{P}(X|Y=1))$ measures the strength of the causal influence. A small value of $|p_0 - p_1|$ indicates weak causal interaction, whereas a large one means the causal influence is strong. Accordingly, we consider two different data settings: *Setting-1: Weak Causation* with $|p_0 - p_1| \in [0.1, 0.2]$ and *Setting-2: Strong Causation* with $|p_0 - p_1| \in [0.8, 0.9]$. To remove the effect of randomness,

we repeat the generation process 20 times. For each time, we generate 100 replications under $\mathbb{H}_0$ and $\mathbb{H}_1$, respectively, to record the type I error rate and recall.

**Comparsion with baselines.** We report the performance of our method and the baseline approaches in Fig. 1. As shown, in *Setting-1: Weak Causation*, our method outperforms the baselines in recall, and achieving competitve type I error rate, indicating effective causal discovery with controlled false positives; wheras in *Setting-2: Strong Causation*, the performance of our method and the baselines is similar. To interpret these results, the advantage of our method in Setting-1 largely stems from the use of the $\mathbb{P}_{DC}$ as the optimization objective, which facilitates a more efficient collection of decisive and correct evidence (see columns (c) and (d)), leading to a better intervention strategy. On the other hand, for Setting-2, the causal relationship is strong and therefore can be identified without the need for refined interventions. Consequently, information gain-based meth-

---

[1]Please refer to Appx. D.4 for results on continuous variables.

*Table 1.* Results under different observational sample sizes $|\mathcal{D}_{\mathrm{obs}}|$

| $|\mathcal{D}_{\mathrm{obs}}|$ | Setting 1 | | Setting 2 | |
|---|---|---|---|---|
| | Type I Error | Recall | Type I Error | Recall |
| 800 | $0.025_{\pm.015}$ | $0.607_{\pm.047}$ | $0.000_{\pm.00}$ | $1.000_{\pm.00}$ |
| 1000 | $0.0385_{\pm.022}$ | $0.627_{\pm.044}$ | $0.001_{\pm.00}$ | $1.000_{\pm.00}$ |
| 1200 | $0.0395_{\pm.013}$ | $0.623_{\pm.059}$ | $0.000_{\pm.00}$ | $1.000_{\pm.00}$ |

ods (Tigas et al., 2022; Toth et al., 2022) can also achieve results similar to ours.

**Ablative study.** We conduct an ablation study on the multi-step optimization algorithm introduced in Sec. 4.3 and present the results in Fig. 2. As we can see, the sequential optimization algorithm significantly enhances the performance of our method, particularly in Setting-1 where the causal relationship is more challenging to identify. This result demonstrates that our sequential optimization scheme can effectively balance the short-term and long-term benefits, leading the optimization of our $\mathbb{P}_{\mathrm{DC}}$ objective towards the optimal solution.

**Influence of the observational sample sizes $|\mathcal{D}_{\mathrm{obs}}|$.** In Tab. 1, we report the type I error rate and recall under different observation sample sizes $|\mathcal{D}_{\mathrm{obs}}| = \{800, 1000, 1200\}$. As shown, our method works consistently well under different observational data settings. This result demonstrates that our method is robust to changes in hyperparameters, consistently achieving reliable causal identification.

## 5.2. Tree-structured causal graph learning

**Data generation.** We consider tree graphs with 30 vertices. To generate the data, we first randomly sample 200 trees from the space of all possible trees. Next, we select a root vertex and generate its value by sampling from $\mathrm{Bern}(0.5)$. For the remaining vertices, we sample according to the following distributions: $\mathbb{P}(X_i|\mathrm{PA}_i = 0) \sim \mathrm{Bern}(\epsilon), \mathbb{P}(X_i|\mathrm{PA}_i = 1) \sim \mathrm{Bern}(1-\epsilon)$, where $\mathrm{PA}_i$ represents the parent of $X_i$ in the tree. For each vertex, we randomly select $\epsilon$ from the interval $[\delta, 0.5 - \delta]$. We explore five different choices of $\delta \in \{0.05, 0.1, 0.15, 0.2, 0.25\}$.

**Results analysis.** In Fig. 3, we show the averaged number of interventions required when applying (Greenewald et al., 2019) and our method to recover the tree graph. As we can see, our method demonstrates consistent advantage over baselines under all settings of $\delta$. Particularly, in the most challenging scenario with $\delta = 0.05$, our method recover the graph with averagely 140 interventions, which is about 10% fewer than the information-theoretic methods proposed by (Tigas et al., 2022; Toth et al., 2022). These results highlight the effectiveness of our Bayes factor objective in optimizing reliable causal discovery, as well as the efficacy of our sequential optimization algorithm in identifying the optimal intervention values.

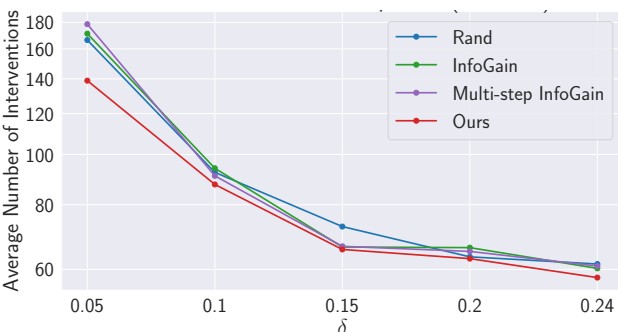

*Figure 3.* Results on tree causal discovery. The Y-axis is the average number of interventions conducted when using (Greenewald et al., 2019) and our method to recover the whole tree. The X-axis represents different choices of the parameter $\delta$. Please refer to Appx. D.5 for variance analysis.

## 5.3. Causal reasoning in embodied AI

In this section, we consider a switch-light reasoning task to illustrate the usage of our method in causal Embodied AI.

**Experimental setup.** We develop a simulation environment based on TongSim (Peng et al., 2024), which includes an embodied agent, two switches ($S_1$ and $S_2$), and two lights ($L_1$ and $L_2$). The agent is tasked to identify which switch controls which light through active intervention. The data of switch activation and lighting status is generated using Bernouli distributions $\mathbb{P}(L_i|S_j = 0) \sim \mathrm{Bern}(p_0)$ and $\mathbb{P}(L_i|S_j = 1) \sim \mathrm{Bern}(p_1)$ for $i, j = 1, 2$. We select the parameter $|p_0 - p_1| = \mathrm{TV}(\mathbb{P}(L_i|S_j = 0), \mathbb{P}(L_i|S_j = 1))$ from the set $\{(0.2, 0.3), (0.4, 0.5), (0.6, 0.7), (0.8, 0.9)\}$ for various strength of causal influence. To further increase the complexity, we introduce a latent confounder between the switch activations, making the results unidentifiable based solely on observational data. To remove the effect of randomness, we repeat the generation process 100 times. The environment and its underlying causal structures are illustrated in Fig. 4.

**Result analysis.** Tab. 2 shows the average number of interventions needed to recover the underlying causal structures. As we can see, our method demonstrates competitive performance across various settings of $|p_0 - p_1|$. Notably, the advantage is particularly pronounced when $|p_0 - p_1|$ is small and the causal influence is weak, which aligns with observations from the bivariate causal discovery experiment. These results underscore the applicability of our method to causal reasoning tasks in Embodied AI.

## 6. Conclusions and limitations

In this paper, we propose a novel Bayesian framework for active causal discovery. We formulated an objective based on Bayes factors and the probability of obtaining decisive

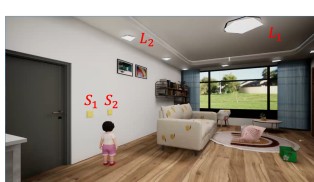

$$\mathbb{H}_0 : \quad S_1 \leftarrow\!\!\!\shortrightarrow S_2$$
$$\qquad\qquad \downarrow \qquad \downarrow$$
$$\qquad\qquad L_2 \qquad L_1$$

$$\mathbb{H}_1 : \quad S_1 \leftarrow\!\!\!\shortrightarrow S_2$$
$$\qquad\qquad \downarrow \qquad \downarrow$$
$$\qquad\qquad L_1 \qquad L_2$$

*Figure 4.* The simulation environment for the switch-light reasoning task. The agent is required to identify which swich ($S_1$ and $S_2$) controls which light ($L_1$ and $L_2$) though active exploration. The causal graphs under $\mathbb{H}_0$ and $\mathbb{H}_1$ are shown on the right.

*Table 2.* Average number of interventions required in the switch-light reasoning task. Best results are **boldfaced**.

| Methods | Range of $|p_0 - p_1|$ | | | |
|---|---|---|---|---|
| | (0.2, 0.3) | (0.4, 0.5) | (0.6, 0.7) | (0.8, 0.9) |
| Rand | $46.15_{\pm14.5}$ | $14.77_{\pm2.7}$ | $7.44_{\pm1.3}$ | $4.00_{\pm0.7}$ |
| InfoGain | $33.51_{\pm13.9}$ | $9.97_{\pm3.9}$ | $5.05_{\pm1.7}$ | $\mathbf{2.60_{\pm0.8}}$ |
| Multi-step InfoGain | $32.52_{\pm14.3}$ | $\mathbf{9.66_{\pm3.6}}$ | $4.99_{\pm1.6}$ | $2.63_{\pm0.7}$ |
| Ours | $\mathbf{32.11_{\pm13.2}}$ | $9.88_{\pm3.3}$ | $\mathbf{4.89_{\pm1.5}}$ | $2.61_{\pm0.7}$ |

and correct evidence for causal relationships under interventions. To optimize this objective, we present an efficient sequential optimization algorithm grounded in dynamic programming. The effectiveness of our method is demonstrated through superior performance in bivariate causal discovery and tree-structured causal graph learning tasks.

**Limitation and Future works.** While our method can effectively identify causal relationships in the bivariate case, extending it to multivariate graphs can be non-trivial, since we currently rely on the identifiability of $p\{y|\mathrm{do}(x)\}$ for optimization. To resolve this problem, we will develop algorithms that simultaneously estimate the interventional distribution and compute the Bayes factors from the interventional samples for a dedicated solution. Moreover, we are also interested in applying our method to various causal reasoning and intuitive physics tasks in Embodied AI.

## Acknowledgements

This work was supported by the National Science and Technology Major Project (2022ZD0114904) and NSFC-6247070125.

## Impact Statement

This paper presents work whose goal is to advance the field of Machine Learning. There are many potential societal consequences of our work, none which we feel must be specifically highlighted here.

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

# Appendix

# A. The connection between $\mathbb{P}_{\text{DC}}$ and information gain

Here we provide a detailed analysis of the relationship between our objective function and the information gain criterion. The mutual information between the hypothesis $\mathbb{H}$ and new observation $Y$ given intervention $X = x$ can be expressed as:

$$I(Y; \mathbb{H} \mid X = x) = \sum_h \int_{\mathcal{Y}} p\{y, \mathbb{H} = h \mid do(X = x)\} \log \frac{p\{y, \mathbb{H} = h \mid do(X = x)\}}{p\{y \mid do(X = x), \mathbb{H} = h\}} \mathrm{d}y$$
$$= \sum_h \int_{\mathcal{Y}} p\{y \mid do(X = x), \mathbb{H} = h\} \mathbb{P}(\mathbb{H}) \log \frac{p\{y, \mathbb{H} = h \mid do(X = x)\}}{p\{y \mid do(X = x), \mathbb{H} = h\}} \mathrm{d}y \tag{9}$$

For the term $p\{y, \mathbb{H} = h \mid do(X = x)\}/p\{y \mid do(X = x), \mathbb{H} = h\}$ when $h = \mathbb{H}_0$, we can derive:

$$\frac{p\{y, \mathbb{H}_0 \mid do(X = x)\}}{p\{y \mid do(X = x), \mathbb{H}_0\}} = \frac{p\{y \mid do(X = x), \mathbb{H}_0\}}{\sum_{h'} p\{y \mid do(X = x), h'\} p\{h'\}}$$
$$= \frac{p\{y \mid do(X = x), \mathbb{H}_0\}}{p\{y \mid do(X = x), \mathbb{H}_0\} p\{\mathbb{H}_0\} + p\{y \mid do(X = x), \mathbb{H}_1\} p\{\mathbb{H}_1\}} \tag{10}$$
$$= \frac{\text{BF}_{01}(\{(x, y_{\text{new}})\})}{\text{BF}_{01}(\{(x, y_{\text{new}})\}) p\{\mathbb{H}_0\} + p\{\mathbb{H}_1\}}$$

Similarly, when $h = \mathbb{H}_1$, we can obtain:

$$\frac{p\{y, \mathbb{H}_1 \mid do(X = x)\}}{p\{y \mid do(X = x), \mathbb{H}_1\}} = \frac{1}{\text{BF}_{01}(\{(x, y_{\text{new}})\}) p\{\mathbb{H}_0\} + p\{\mathbb{H}_1\}} \tag{11}$$

Therefore:

$$I(Y; \mathbb{H} \mid X = x) = \int_{\mathcal{Y}} p\{\mathbb{H}_0\} \log \frac{\text{BF}_{01}(\{(x, y_{\text{new}})\})}{\text{BF}_{01}(\{(x, y_{\text{new}})\}) p\{\mathbb{H}_0\} + p\{\mathbb{H}_1\}} p\{Y = y \mid do(X = x), \mathbb{H}_0\} \mathrm{d}y$$
$$+ \int_{\mathcal{Y}} p\{\mathbb{H}_1\} \log \frac{1}{\text{BF}_{01}(\{(x, y_{\text{new}})\}) p\{\mathbb{H}_0\} + p\{\mathbb{H}_1\}} p\{Y = y \mid do(X = x), \mathbb{H}_1\} \mathrm{d}y \tag{12}$$

Comparing the summands in equations 5 and above, we can observe that both are monotonic functions of $\text{BF}_{01}(\{(x, y_{\text{new}})\})$. When $\text{BF}_{01}(\{(x, y_{\text{new}})\})$ increases, both the indicator function in $\mathbb{P}_{\text{DC}}$ and the logarithm term in information gain increase, indicating these objectives are aligned in preferring interventions that yield high Bayes factors when $\mathbb{H}_0$ is true, and low Bayes factors when $\mathbb{H}_1$ is true.

# B. Multi-step optimization

### B.1. Proof of symmetry property

Here we provide a detailed proof that $\mathbb{P}_{\text{DC}}(x_{1:k})$ is symmetric in its arguments $(x_1, \cdots, x_k)$ for any fixed $k$. This symmetry property is crucial for reducing the computational complexity of our optimization problem.

**Proposition B.1.** *For any fixed $k$, $\mathbb{P}_{\text{DC}}(x_{1:k})$ is symmetric in its arguments $(x_1, \cdots, x_k)$.*

*Proof.* By definition of $\mathbb{P}_{\text{DC}}$ (Equation (6)), we have:

$$\mathbb{P}_{\text{DC}}(x_{1:k}) = \mathbb{P}\{\text{BF}_{01}(\{\overline{x}_1, \overline{y}_1\} \cup \mathcal{D}_{\text{fut}}) > k_0 \mid \mathbb{H}_0\} \mathbb{P}(\mathbb{H}_0) + \mathbb{P}\{\text{BF}_{01}(\{\overline{x}_1, \overline{y}_1\} \cup \mathcal{D}_{\text{fut}}) < k_1 \mid \mathbb{H}_1\} \mathbb{P}(\mathbb{H}_1)$$
$$= \sum_{y_{1:k} \in \mathcal{Y}^k} \mathbb{P}\{y_{1:k} \mid do(x_{1:k}), \mathbb{H}_0\} \mathbb{P}(\mathbb{H}_0) \mathbb{I}\{\text{BF}_{01}(\{\overline{x}_1, \overline{y}_1\} \cup \mathcal{D}_{\text{fut}}) > k_0\}$$
$$+ \sum_{y_{1:k} \in \mathcal{Y}^k} \mathbb{P}\{y_{1:k} \mid do(x_{1:k}), \mathbb{H}_1\} \mathbb{P}(\mathbb{H}_1) \mathbb{I}\{\text{BF}_{01}(\{\overline{x}_1, \overline{y}_1\} \cup \mathcal{D}_{\text{fut}}) < k_1\} \tag{13}$$

For each hypothesis $\mathbb{H}_j$, the likelihood term can be expressed as:

$$\mathbb{P}\{y_{1:k} \mid do(x_{1:k}), \mathbb{H}_j\} = \int \prod_{i=1}^{k} \mathbb{P}\{y_i \mid do(x_i), \theta, \mathbb{H}_j\} \mathbb{P}\{\theta \mid \mathbb{H}_j\} d\theta \qquad (14)$$

The Bayes factor is the ratio of these likelihoods:

$$\mathrm{BF}_{01}(\{\overline{x}_1, \overline{y}_1\} \cup \mathcal{D}_{\mathrm{fut}}) = \frac{\int \prod_{i=1}^{k} \mathbb{P}\{y_i \mid do(x_i), \theta, \mathbb{H}_0\} \mathbb{P}\{\theta \mid \mathbb{H}_0\} d\theta}{\int \prod_{i=1}^{k} \mathbb{P}\{y_i \mid do(x_i), \theta, \mathbb{H}_1\} \mathbb{P}\{\theta \mid \mathbb{H}_1\} d\theta} \qquad (15)$$

Both the likelihood terms and the Bayes factor are invariant under any permutation of the indices of $(y_1, \cdots, y_k)$ due to the product form inside the integrals. By Fubini's theorem, we can interchange the order of summation over $y_{1:k}$ in the original expression. Since both the likelihoods and the decision criteria (through the Bayes factor) are invariant under permutations of corresponding $(x_i, y_i)$ pairs, $\mathbb{P}_{\mathrm{DC}}(x_{1:k})$ must be symmetric in $(x_1, \cdots, x_k)$. $\qquad \square$

This symmetry property leads to significant computational savings in the main optimization problem. Rather than evaluating all possible ordered sequences of interventions, we only need to consider unique combinations of intervention values, reducing the complexity from $O(|\mathcal{X}|^k)$ to $O(k^{|\mathcal{X}|})$.

### B.2. Proof of count-dependent only property

To develop an efficient optimization algorithm, we first prove that our k-step optimization decisions depend only on the aggregate statistics of previous interventions. This property enables us to solve the problem recursively using dynamic programming.

**Proposition B.2.** *For the sequential optimization problem in Equation 7, the optimal policy $\pi_t^* : \mathcal{X}^{t-1} \to \mathcal{X}$ at step t depends only on the aggregate statistics of previous interventions:*

$$\pi_t^* (x_{1:t-1}) = f_t \left( \sum_{i=1}^{t-1} \mathbf{e}_{x_i} \right)$$

*where $\mathbf{e}_{x_i}$ is the unit vector corresponding to intervention $x_i$, and $f_t : \mathbb{N}^{|\mathcal{X}|} \to \mathcal{X}$ is some function.*

*Proof.* Let $\mathbf{n}_t \in \mathbb{N}^{|\mathcal{X}|}$ denote the vector of counts for each intervention value in $\mathcal{X}$ up to step t, where the i-th element represents how many times the i-th intervention value has been used in the first t steps. Our objective function is to maximize $\sum_{i=1}^{k} \mathbb{P}_{\mathrm{DC}}(x_{1:i})$. By the symmetry property (Proposition 4.1), $\mathbb{P}_{\mathrm{DC}}(x_{1:i}) = \mathbb{P}_{\mathrm{DC}}(\mathbf{n}_i)$, meaning the value at each step depends only on the count vector rather than the specific sequence of interventions.

We can then transform the optimization problem as follows:

$$\max_{\mathbf{x}} \sum_{k=1}^{K} \gamma^k \mathbb{P}_{\mathrm{DC}}(x_{1:k})$$

$$\iff \max_{\text{valid } \mathbf{n}_1, \cdots, \mathbf{n}_K} \sum_{k=1}^{K} \gamma^k \mathbb{P}_{\mathrm{DC}}(\mathbf{n}_k)$$

$$\iff \max_{\text{valid } \mathbf{n}_K} \left\{ \max_{\text{valid } \mathbf{n}_1, \cdots, \mathbf{n}_{K-1}} \sum_{k=1}^{K-1} \gamma^k \mathbb{P}_{\mathrm{DC}}(\mathbf{n}_k) \right\} + \gamma^K \mathbb{P}_{\mathrm{DC}}(\mathbf{n}_K) \qquad (16)$$

Here, a count vector $\mathbf{n}_k$ is valid if there exists an intervention sequence $x_{1:k}$ that generates it, i.e., if there exists some $x_k$ such that $\mathbf{n}_k = \mathbf{n}_{k-1} + \mathbf{e}_{x_k}$. This decomposition reveals that at each step we need to: **(i)** track the current count vector $\mathbf{n}_k$, **(ii)** choose the next intervention $x_k$ that generates a valid next count vector, and **(iii)** solve the subproblem for the previous $k-1$ steps. This structure suggests maintaining value functions $J_k(\mathbf{n}_k)$ and policies $\pi_k(\mathbf{n}_k)$ for each possible count vector at each step, which can be computed recursively through dynamic programming. $\qquad \square$

## B.3. Estimation and optimization for continuous interventions

In the continuous intervention setting, estimating interventional distributions and computing likelihoods requires careful consideration of the continuous parameter space. Here we detail our approach for implementation.

## B.4. Estimation of Interventional Distributions

For continuous interventions, we first estimate the parameters of the interventional distributions under each hypothesis using maximum likelihood estimation:

Under $\mathbb{H}_0$ where $X$ has no causal effect on $Y$, the interventional distribution reduces to the marginal distribution:

$$\mathbb{P}\{y \mid \text{do}(X = x), \theta, \mathbb{H}_0\} = \mathbb{P}\{y \mid \theta\} \tag{17}$$

Under $\mathbb{H}_1$ where $X$ causes $Y$, the interventional distribution equals the conditional distribution:

$$\mathbb{P}\{y \mid \text{do}(X = x), \theta, \mathbb{H}_1\} = \mathbb{P}\{y \mid x, \theta\} \tag{18}$$

The maximum likelihood estimates $\hat{\theta}$ for each hypothesis are obtained by:

$$\hat{\theta}_{\mathbb{H}} = \arg \max_{\theta} \sum_{i=1}^{N_{\text{obs}}} \log \mathbb{P}\{y_i \mid x_i, \theta, \mathbb{H}\} \tag{19}$$

## B.5. Parameter Prior and Likelihood Computation

Following (Kass & Raftery, 1995), we employ a data-dependent parameter prior centered around the MLE estimate. For each hypothesis $\mathbb{H}$, we approximate the parameter posterior as a Gaussian distribution:

$$\mathbb{P}\{\theta \mid \mathcal{D}_m, \mathbb{H}\} \approx \mathcal{N}(\hat{\theta}, \hat{\Sigma}) \tag{20}$$

where $\hat{\Sigma}$ is the inverse Hessian of the negative log-likelihood at $\hat{\theta}$.

The marginal likelihood is then approximated using Laplace's method:

$$\mathbb{P}\{y \mid \text{do}(X = x), \mathbb{H}\} \approx (2\pi)^{d/2} \mathbb{P}\{y \mid \text{do}(X = x), \hat{\theta}, \mathbb{H}\} |\hat{\Sigma}|^{1/2} \mathbb{P}\{\hat{\theta} \mid \mathbb{H}\} \tag{21}$$

In practice, computing the Hessian for all data points in each likelihood evaluation can be computationally intensive. An alternative is to use an empirical Bayes approach with a point mass prior ($\hat{\Sigma} \to 0$), which reduces the computation to a likelihood ratio test while maintaining computational efficiency.

## B.6. Smooth Approximation for Gradient-based Optimization

To enable gradient-based optimization in continuous intervention space, we need to address the non-differentiability of the indicator functions in $\mathbb{P}_{\text{DC}}$. We introduce a smooth approximation using the exponential function:

$$H_\beta(x) = \begin{cases} \exp(-x/\beta) & \text{if } x < 0 \\ 1 & \text{if } x \geq 0 \end{cases} \tag{22}$$

This can be expressed compactly using the ReLU function:

$$H_\beta(x) = \exp(-\text{ReLU}(-x)/\beta) \tag{23}$$

Using this approximation, we can first rewrite the single-step $\mathbb{P}_{\text{DC}}$ in a differentiable form:

$$\begin{aligned}
\mathbb{P}_{\text{DC}}(\mathcal{D}_m, x) \approx{} & \mathbb{E}_{y \sim \hat{m}_0} \left\{ \exp(-\text{ReLU}(k_0 - \text{BF}_{01}(\mathcal{D}_m \cup \{(x, y)\}))/\beta) \right\} \mathbb{P}(\mathbb{H}_0 \mid \mathcal{D}_m) \\
& + \mathbb{E}_{y \sim \hat{m}_1} \left\{ \exp(-\text{ReLU}(\text{BF}_{01}(\mathcal{D}_m \cup \{(x, y)\}) - k_1)/\beta) \right\} \mathbb{P}(\mathbb{H}_1 \mid \mathcal{D}_m)
\end{aligned} \tag{24}$$

The expectations are approximated using Monte Carlo sampling:

$$\hat{\mathbb{P}}_{\text{DC}} = \frac{1}{N} \sum_{i=1}^{N} \exp\left\{-\text{ReLU}(k_0 - \text{BF}_{01}(\mathcal{D}_m \cup \{(x, y_i^0)\}))/\beta\right\} \mathbb{P}(\mathbb{H}_0 \mid \mathcal{D}_m)$$
$$+ \frac{1}{N} \sum_{i=1}^{N} \exp\left\{-\text{ReLU}(\text{BF}_{01}(\mathcal{D}_m \cup \{(x, y_i^1)\}) - k_1)/\beta\right\} \mathbb{P}(\mathbb{H}_1 \mid \mathcal{D}_m)$$

(25)

Similarly, for the multi-step case, we can write:

$$\mathbb{P}_{\text{DC}}(\overline{x}_{1:K}) \approx \mathbb{E}_{y \sim \hat{m}_0} \left\{\exp(-\text{ReLU}(k_0 - \text{BF}_{01}(\mathcal{D}_m \cup \{(\overline{x}_1, \overline{y}_1)\} \cup \mathcal{D}_{\text{fut}}))/\beta)\right\} \mathbb{P}(\mathbb{H}_0 \mid \mathcal{D}_m)$$
$$+ \mathbb{E}_{y \sim \hat{m}_1} \left\{\exp(-\text{ReLU}(\text{BF}_{01}(\mathcal{D}_m \cup \{(\overline{x}_1, \overline{y}_1)\} \cup \mathcal{D}_{\text{fut}}) - k_1)/\beta)\right\} \mathbb{P}(\mathbb{H}_1 \mid \mathcal{D}_m)$$

(26)

where $\mathcal{D}_{\text{fut}} = \{(\overline{x}_2, \overline{y}_2), ..., (\overline{x}_K, \overline{y}_K)\}$ represents data of the next $K-1$ steps.

This differentiable approximation enables the use of gradient-based optimization methods to find the optimal intervention values. The smoothing parameter $\beta$ controls the trade-off between approximation accuracy and optimization stability.

## C. Application to multivariate causal discovery

To illustrate both the applicability and limitations of our method in general graphs, consider a three-node complete graph (as shown in Figure 5). Figure 6 shows all possible configurations under $\mathbb{H}_0 : X \leftarrow Y$, and Figure 7 shows configurations under $\mathbb{H}_1 : X \rightarrow Y$.

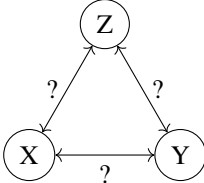

*Figure 5.* A complete three-node graph with uncertain edge directions

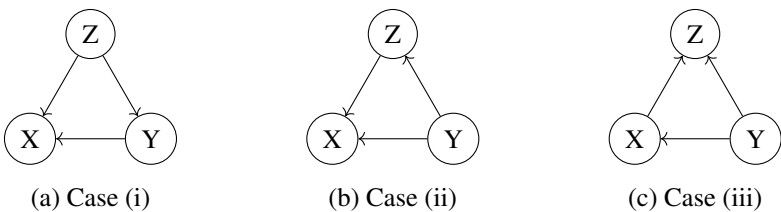

(a) Case (i)               (b) Case (ii)               (c) Case (iii)

*Figure 6.* Possible configurations under $\mathbb{H}_0 : X \leftarrow Y$

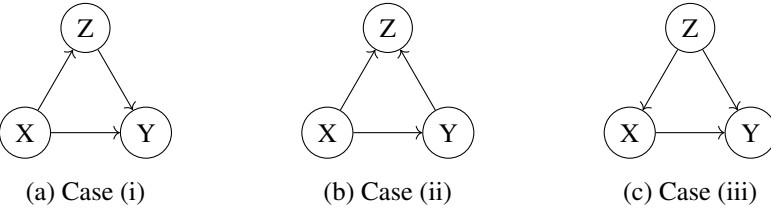

(a) Case (i)               (b) Case (ii)               (c) Case (iii)

*Figure 7.* Possible configurations under $\mathbb{H}_1 : X \rightarrow Y$

When testing the causal direction between X and Y in this setting, under $\mathbb{H}_0 : X \leftarrow Y$, the possible configurations are shown in Figure 6. For all these cases, $\mathbb{P}\{Y \mid \text{do}(X), \mathbb{H}_0\} = \mathbb{P}\{Y\}$ since X has no causal effect on Y.

Under $\mathbb{H}_1 : X \to Y$, we have the configurations shown in Figure 7. The interventional distributions differ across these cases: for cases (a) and (b), $\mathbb{P}\{Y \mid \mathrm{do}(X), \mathbb{H}_1\} = \mathbb{P}\{Y \mid X\}$, while for case (c), $\mathbb{P}\{Y \mid \mathrm{do}(X), \mathbb{H}_1\} = \int \mathbb{P}\{Y \mid X, Z = z\}\mathbb{P}\{Z = z\}\mathrm{d}z$.

However, when certain structural information is known (e.g., if we know $X \leftarrow Z \to Y$ exists), the interventional distributions become uniquely identifiable, corresponding to case (c) in Figure 7. In such cases, our method directly applies.

# D. Experiments

### D.1. Bayes Factor classification

The Bayes factor can be classified into different categories of evidence strength (Jeffreys, 1961; Schönbrodt & Wagenmakers, 2017), which helps interpret its value in supporting $\mathbb{H}_0$ or $\mathbb{H}_1$. Table 3 presents these classifications:

*Table 3.* Classification for the evidence levels of the Bayes factor $\mathrm{BF}_{01}$ (from (Schönbrodt & Wagenmakers, 2017) adapted from (Jeffreys, 1961)).

| Bayes factor | Evidence Level |
|:---:|:---:|
| $> 100$ | Extreme evidence for $\mathbb{H}_0$ |
| $30 - 100$ | Very strong evidence for $\mathbb{H}_0$ |
| $10 - 30$ | Strong evidence for $\mathbb{H}_0$ |
| $3 - 10$ | Moderate evidence for $\mathbb{H}_0$ |
| $1 - 3$ | Anecdotal evidence for $\mathbb{H}_0$ |
| $1$ | No evidence |
| 1/3 - 1 | Anecdotal evidence for $\mathbb{H}_1$ |
| 1/10 - 1/3 | Moderate evidence for $\mathbb{H}_1$ |
| 1/30 - 1/10 | Strong evidence for $\mathbb{H}_1$ |
| 1/100 - 1/30 | Very strong evidence for $\mathbb{H}_1$ |
| $< 1/100$ | Extreme evidence for $\mathbb{H}_1$ |

This classification provides a systematic framework for interpreting Bayes factors, with values further from 1 indicating stronger evidence for the respective hypothesis. The reciprocal nature of the classification (e.g., $\mathrm{BF}_{01} > 100$ and $\mathrm{BF}_{01} < 1/100$ representing extreme evidence for opposing hypotheses) reflects the symmetry in evidence interpretation.

### D.2. Additional results on the discrete setting with $k_0 = 10$

This appendix presents comprehensive performance results across the full range of total variation (TV) distances from 0 to 1.0, examining how the choice of $|p_1 - p_0|$ described in Section 5 affects both the recall rate and Type I error control. For each TV distance range (spanning intervals of 0.1), we show recall rates under $\mathbb{H}_1$ and Type I error rates under $\mathbb{H}_0$, presented in consecutive columns.

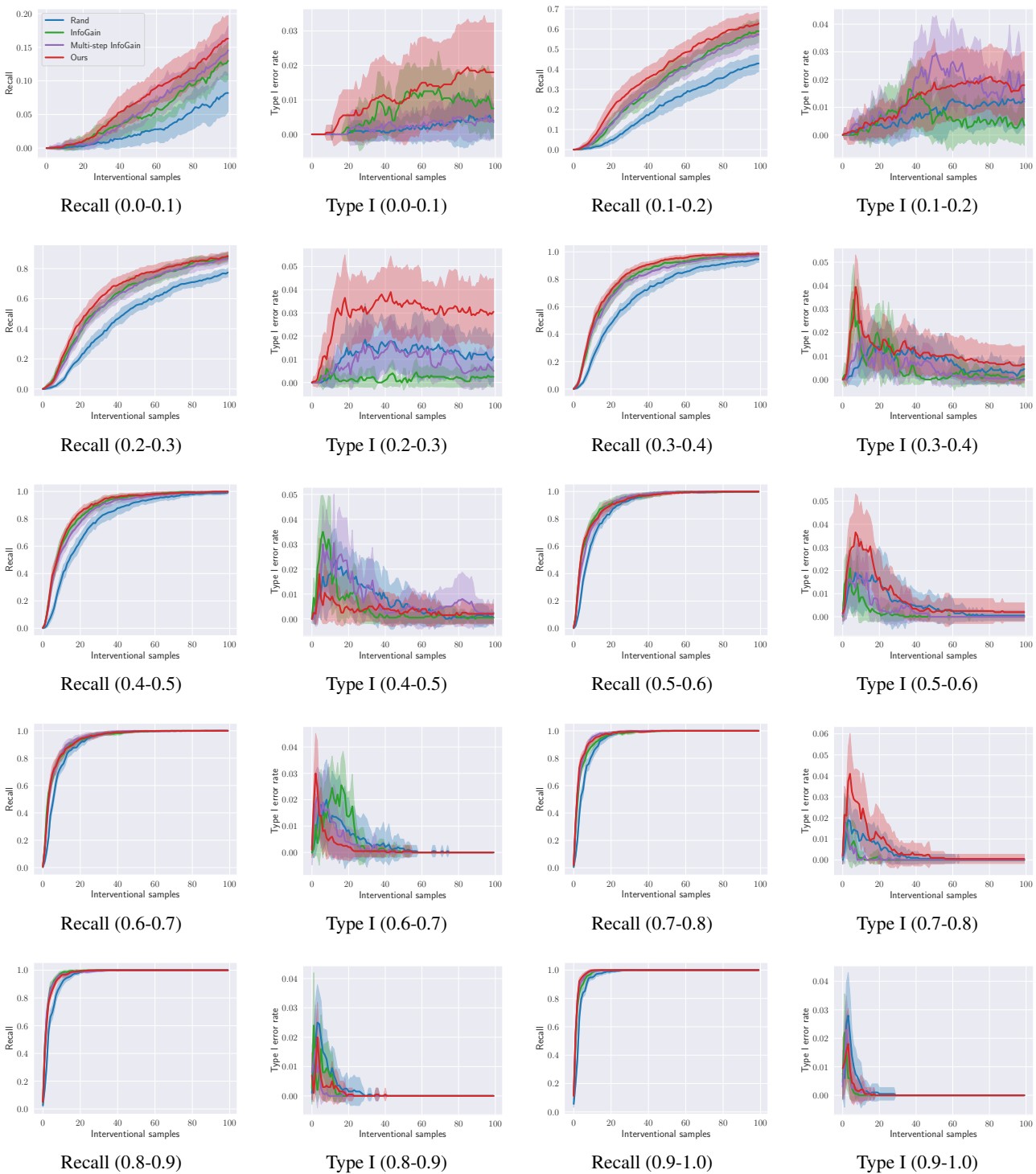

*Figure 8.* Performance evaluation showing recall rates and Type I error rates for different causal strengthness (evalueted by the total variation (TV) distance ranges between $\mathbb{P}\{Y \mid \mathrm{do}(X = 0)\}$ and $\mathbb{P}\{Y \mid \mathrm{do}(X = 1)\}$

These results provide a complete picture of method performance across different distributional distances, complementing the focused analysis of representative TV ranges presented in the main text. The figures maintain a consistent layout: each row shows two pairs of metrics - recall and Type I error rates for consecutive TV distance ranges, allowing direct comparison of performance as the distributional similarity varies from highly similar (TV range 0-0.1) to highly distinct (TV range 0.9-1.0).

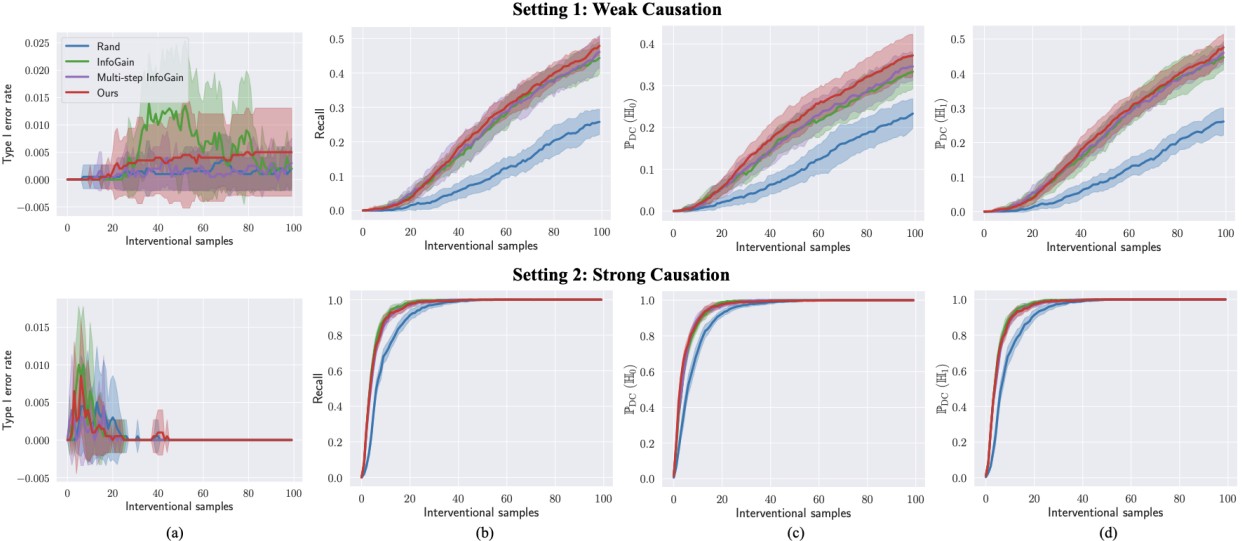

*Figure 9.* Results on bivariate causal discovery with $k_0 = 30$ (very strong evidence). The two rows presents results in two different settings. Columns (a)-(d) report the type I error rate, recall, $\mathbb{P}_{\text{DC}}$ under $\mathbb{H}_0$, and $\mathbb{P}_{\text{DC}}$ under $\mathbb{H}_1$, respectively, in relation to the number of intervention samples.

The paired layout facilitates examination of how performance evolves across adjacent TV ranges, with each row capturing the transition between consecutive intervals.

### D.3. Performance evaluation with different evidence thresholds

Following established Bayesian evidence interpretation frameworks (Jeffreys, 1961; Schönbrodt & Wagenmakers, 2018), we conducted additional experiments with more stringent evidence thresholds to verify the robustness of our proposed method. This subsection presents results for $k_0 = 30$ (very strong evidence) and $k_0 = 100$ (extreme evidence), complementing the $k_0 = 10$ results shown above. These experiments were conducted in response to reviewer feedback and demonstrate the utility of our methods across different evidence strength requirements.

**Results with stringent evidence thresholds.** We report the performance under more stringent evidence requirements in Fig. 9 and Fig. 10. Our method consistently outperforms baselines in recall while maintaining competitive type I error rates across both $k_0 = 30$ and $k_0 = 100$ settings. The advantage stems from our $\mathbb{P}_{\text{DC}}$-based optimization that efficiently collects decisive evidence. Importantly, the superior performance of our approach remains consistent across different evidence threshold requirements, demonstrating the robustness of the proposed intervention strategy.

### D.4. Experiments on Continuous Settings

**Experimental Setup** For continuous intervention variables, we employ the Adam optimizer (Kingma & Ba, 2014) with cosine learning rate scheduling (Loshchilov & Hutter, 2016) (initial rate 0.1, cosine decaying to 0.001 over 200 steps, total 4000 optimization steps). Following the same rejection criterion as in the discrete case, we maintain $k_0 = \frac{1}{k_1} = 10$ and use $n_s = 1,000$ observational samples with $T = 100$ sequential interventions.

**Data Generation** We evaluate our method using additive noise models (ANMs):

Under $\mathbb{H}_1 : X \rightarrow Y$:

$$Y = f(X) + \varepsilon \tag{27}$$

Under $\mathbb{H}_0 : X \leftarrow Y$:

$$X = f(Y) + \varepsilon \tag{28}$$

Here $f(x) = \tanh(x)$ is the nonlinear link function. To test robustness across different noise distributions, we create a pool

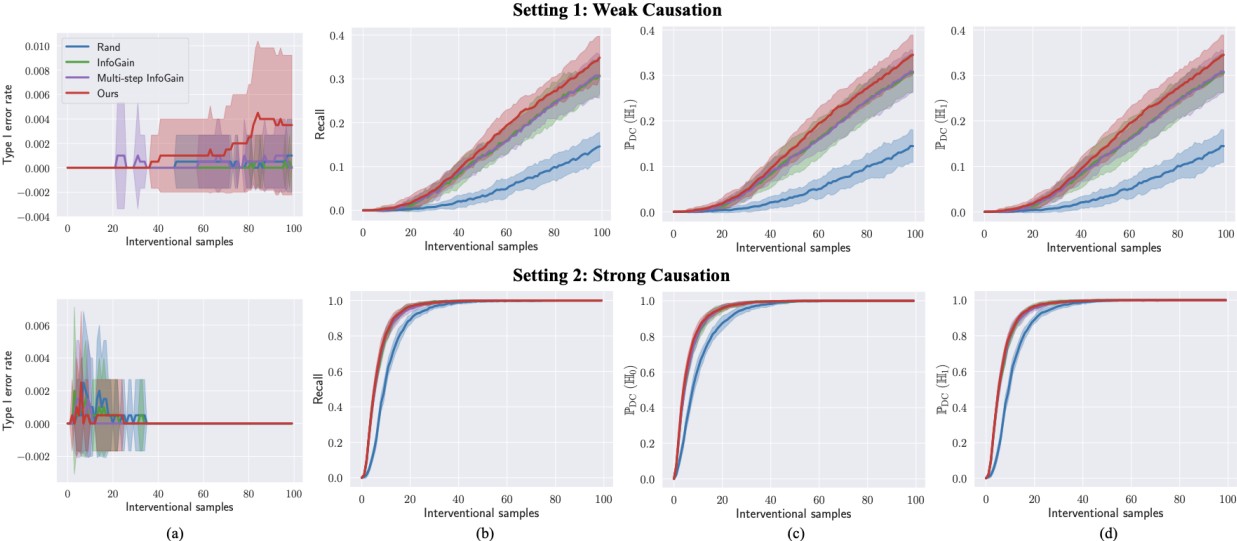

Figure 10. Results on bivariate causal discovery with $k_0 = 100$ (extreme evidence). The two rows presents results in two different settings. Columns (a)-(d) report the type I error rate, recall, $\mathbb{P}_{\mathrm{DC}}$ under $\mathbb{H}_0$, and $\mathbb{P}_{\mathrm{DC}}$ under $\mathbb{H}_1$, respectively, in relation to the number of intervention samples.

of noise models including uniform, normal, Student's t, and two different 4-component Gaussian mixtures:

$$\varepsilon \sim \sum_{i=1}^{4} \pi_i \mathcal{N}(\mu_i, \sigma_i^2), \quad \pi = \frac{1}{8}\mathbf{1} + \frac{3}{4}\mathrm{Softmax}(\mathbf{z}) \tag{29}$$

where $\mathbf{z} \sim \mathcal{N}(0, I_4)$. For each experimental setting, we randomly select one distribution from this pool. This approach allows us to evaluate our method's performance under diverse and complex noise patterns.

**Parameter Estimation**    For each hypothesis $\mathbb{H}$, we model the functional relationship as:

$$f_\theta(x) = a \tanh(bx) \tag{30}$$

where $\theta = (a, b)$ are learnable parameters. The output distribution is modeled using a mixture of 4 Gaussian components to approximate arbitrary distributions flexibly. We jointly learn these structural parameters along with the mixture model parameters (means, variances, and mixing weights) using maximum likelihood estimation on observational data. The optimization is performed using Adam optimizer (Kingma & Ba, 2014) for 10,000 steps with an initial learning rate of 0.1, which is decreased by a factor of 2 at steps 3000, 5000, and 7500.

After learning the model parameters from observational data, we compute the likelihood of interventional samples using the method described in Appendix B.5.

**Results**    The experimental results in Figure 11 demonstrate that all methods achieve comparable performance in the continuous setting. Both recall and Type I error rates show similar performance levels across different approaches, with well-controlled Type I error rates. This convergence can be attributed to the identifiability properties of ANMs, where causal direction can often be determined from observational data alone. The intervention selection strategy thus becomes less critical compared to the discrete case, highlighting how model structure influences the complexity of causal discovery.

### D.5. Variance Analysis for Tree Structure Experiments

To better understand the performance characteristics of different methods, we analyze the distribution of required interventions across 200 random trials, visualized through violin plots partitioned into five groups. While the average performance of random intervention appears competitive in certain settings, the violin plots reveal high variance in its performance distribution. Specifically, the random strategy exhibits a bimodal distribution with a concentrated mass of trials requiring

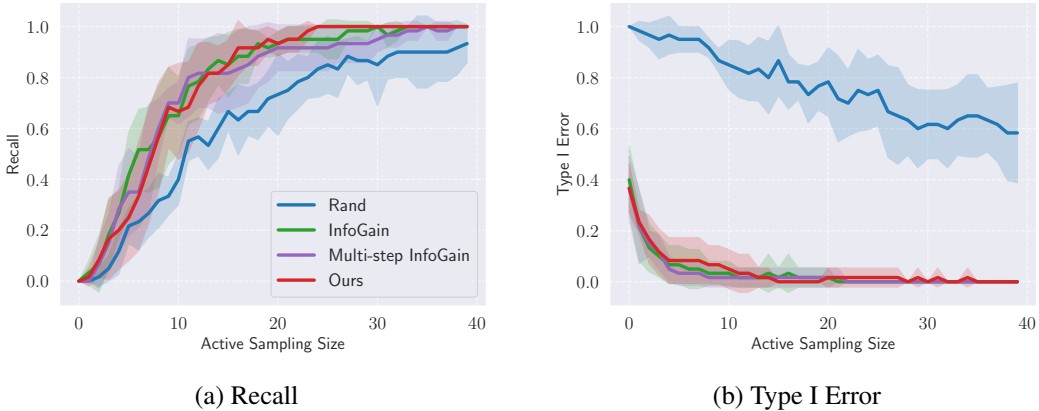

(a) Recall            (b) Type I Error

*Figure 11.* Performance comparison in continuous settings showing recall and Type I error rates. All methods achieve comparable performance with relatively few interventions, suggesting that in the continuous additive noise setting, causal direction can be effectively determined regardless of the intervention strategy.

few interventions and a longer tail of trials requiring significantly more interventions. In contrast, both information gain and multi-step information gain methods demonstrate more uniform distributions, indicating more consistent and predictable performance across trials. This suggests that the seemingly competitive average performance of random intervention is largely driven by a small subset of fortunate trials, rather than reliable systematic efficiency.

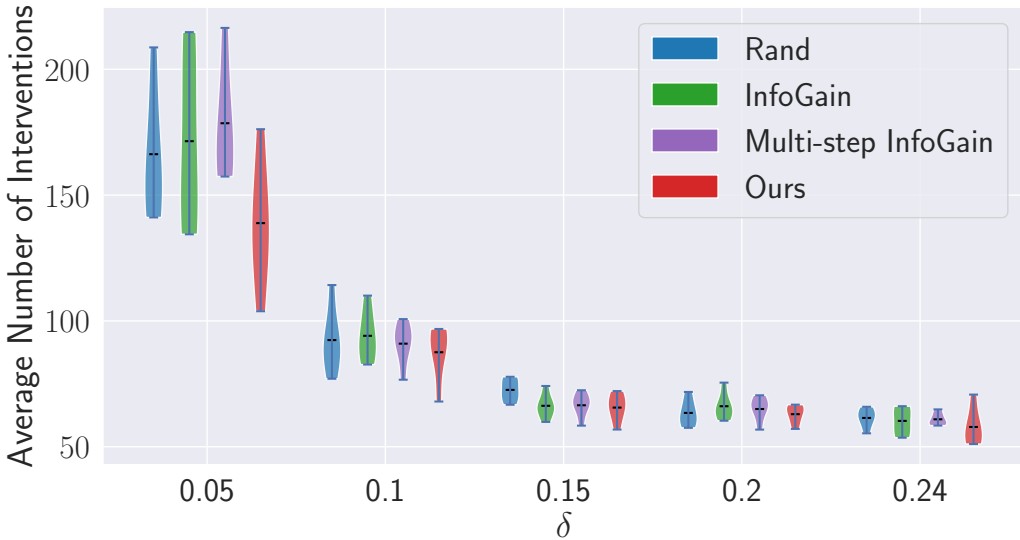

*Figure 12.* Distribution of required interventions across different effect sizes $\varepsilon$, visualized using violin plots. Each violin plot represents the distribution of 200 trials partitioned into five groups.

