# OpenReview forum: "Bayesian Active Learning for Bivariate Causal Discovery"
_ICML.cc/2025/Conference — ICML 2025 poster_

### Official Review · Reviewer_uXGK · 2025-03-06

**Overall Recommendation:** 3

**Summary:**

This paper investigates the problem of identifying the causal direction between two variables, i.e., identify whether x->y or y-> x, through Bayesian active intervention. The paper addresses this problem as a hypothesis testing problem, with a testing statistic called "probability of decisive and correct evidence" ($P_{DC}$) that depends on the Bayes factor (likelihood ratio of the hypothesis over data D). It then proposes an iterative active learning strategy that selects intervention values of X in order to gain most evidence according to the $P_{DC}$ criterion. It also provides a dynamic programming optimization strategy for planning interventions over multiple steps that reduces the combinatorial search space.
The paper then discusses 3 main experiments: (1) binary X, Y with two different causal directions in the form of Bernoulli distributions; (2) tree structured distribution and (3) a simulation environment with switch and light. In all 3 experiment environments, it compares their method with 3 baselines: random, infogain, and multi-step inforgain where it shows sample efficiency of their method over the baselines.

**Claims And Evidence:**

I am not sure whether I agree with the authors that the proposed test statistic $P_{DC}$ is valid for sample selection. From my understanding, the goal of the paper is to give a criteria for actively selecting new data to collect evidence for either direction, that is, each new selection of x_i is dependent on the historical trajectory. However, the statistic $P_{DC}$ depends on the Bayes factor which assumes complete factorization for each x_i, i.e., independence.

**Essential References Not Discussed:**

n.a.

**Experimental Designs Or Analyses:**

The experiment setting seem a bit weak to me, in that all the experiments provided essentially only focus on the case of binary variables and Bernoulli distribution.
Specifically, the paper discusses 3 main experiments: (1) binary X, Y with two different causal directions in the form of Bernoulli distributions; (2) tree structured distribution and (3) a simulation environment with switch and light. In all 3 experiment environments, it compares their method with 3 baselines: random, infogain, and multi-step inforgain. The "strength" of the causation is tuned by the magnitude of the difference in conditional probabilities.

I'm not sure how the embodied AI switch-light reasoning task is more complicated than the binary example in 5.1 and 5.2, given the switches S_1, S_2, and lights L_1, L_2 are just indicator variables. The paper mentions the addition of latent confounder, but how does the confounder affect the dynamic and why does it make the problem harder?

Moreover, in table 2, it is not clear that the paper's method performs better than the other methods, given the performances of the paper's method are inside the confidence set of the baseline performances for every single instance.

**Methods And Evaluation Criteria:**

The paper only discusses two very specific distribution classes for X and Y, for the continuous case, the paper uses Gaussian; and for categorical, it uses Beta-Bernoulli. It is unclear to me if the results generalize to other distributions, e.g. computational issues might arise that requires more sophisticated approximation and optimization techniques.

**Other Comments Or Suggestions:**

n.a.

**Other Strengths And Weaknesses:**

n.a.

**Questions For Authors:**

- The paper shows how to deal with tree-structured graphs. How would you adapt your method if the graph is not a tree but a more general DAG?
- If the distribution $p(y\mid x)$ is misspecified, how does that affect the reliability of $P_{DC}$?
- For experiment section 5.3, I'm not sure how the embodied AI switch-light reasoning task is more complicated than the binary example in 5.1 and 5.2, given the switches S_1, S_2, and lights L_1, L_2 are just indicator variables. The paper mentions the addition of latent confounder, but how does the confounder affect the dynamic and why does it make the problem harder?

**Relation To Broader Scientific Literature:**

The paper designs a Bayesian active learning technique to address the question of distinguishing causal direction of bivariate systems.
The proposed selection criteria uses the idea of hypothesis testing to measure the strength of evidence in favor of the null hypothesis over the alternative.

**Theoretical Claims:**

See above.

---

> ### Author Rebuttal · Authors · 2025-04-01
>
> We appreciate your efforts and feedback regarding our paper. We address your concerns below.
>
> **Validity of PDC for sample selection.** Our framework is valid since the factorization of BF still holds if $x\_i$ depends on the historical data. Intuitively, this is because the dependency only happens to the selection of $\overline{x}$, while BF compares the ratio of two hypotheses that differ only in $p\\{y|\mathrm{do}(x)\\}$.
>
> Specifically, the joint probability $\mathbb{P}\\{\mathcal{D}\_m|\mathbb{H}\_0\\}$ is:
>
> $$\mathbb{P}\\{\mathcal{D}\_m|\mathbb{H}\_0\\} = \mathbb{P}\\{\overline{x}\_1\\}\mathbb{P}\\{\overline{y}\_1|\mathrm{do}(\overline{x}\_1)\\} \Pi\_{i=2} \mathbb{P}\\{\overline{x}\_i|\overline{x}\_{1:i-1},\overline{y}\_{1:i-1},\\}\mathbb{P}\\{\overline{y}\_i|\mathrm{do}(\overline{x}\_i)\\}$$
>
> since $\overline{y}\_i$ only depends on the value of intervened $\overline{x}\_i$ during generation. We can derive a similar factorization for $\mathbb{P}\\{\mathcal{D}\_m|\mathbb{H}\_1\\}$. When computing the Bayes factor, the term $\mathbb{P}\\{\overline{x}\_i|\cdot\\}$ was canceled out by the numerator and denominator, leading to:
>
> $$\mathrm{BF}(\mathcal{D}\_m) = \prod\_{i=1}^m \frac{p\\{\overline{y}\_i|\mathrm{do}(\overline{x}\_i),\mathbb{H}\_0\\}}{p\\{\overline{y}\_i|\mathrm{do}(\overline{x}\_i),\mathbb{H}\_1\\}}$$
>
> The next intervention value $\overline{x}\_{m+1}$ is then optimized based on $\mathcal{D}\_m:=\\{(\overline{x}\_i,\overline{y}\_i\\}$.
>
> In summary, while the selection of the intervention value depends on the historical information, the factorization still holds as we can cancel out these terms.
>
> **Distribution classes and generalizability.** As detailed in Appendix D.3, we do not restrict the model to be Gaussian. We have conducted experiments for uniform, normal, Student's t, and Gaussian mixtures and the results are shown in Figure 9.
>
> **Adapting to general DAGs.** As discussed in lines 283-284, our current version cannot work on general graph since it requires $p\\{y|\mathrm{do}(x)\\}$ to be identifiable from observational data.
>
> A promising solution is to extend $\mathbb{P}\_{\mathrm{DC}}$ as the ratio of maximum likelhood over the equivalence class:
>
> $$\mathrm{BF}\_{01} = \frac{\max\_{G \in \mathcal{G}^{(x,y)}\_0} \mathbb{P}\\{\mathcal{D}| G,\mathrm{do}(x),\mathbb{H}\_0\\}}{\max\_{G \in \mathcal{G}^{(x,y)}\_0} \mathbb{P}\\{\mathcal{D}| G,\mathrm{do}(x),\mathbb{H}\_1\\}}$$
>
> where $\mathcal{G}^{(x,y)}\_0$ (*resp.* $\mathcal{G}^{(x,y)}\_1$) denotes the Markov equivalence class regarding $p\\{y|\mathrm{do}(x)\\}$ under $\mathbb{H}\_0$ (*resp.* $\mathbb{H}\_1$). Since $\mathcal{G}^{(x,y)}\_0$ (*resp.* $\mathcal{G}^{(x,y)}\_1$) encompasses the true graph under $\mathbb{H}\_0$ (*resp.* $\mathbb{H}\_1$), it recovers the true likelihood ratio. We will pursue this direction in the future.
>
> **Effect of distribution misspecification.** If $p\\{y|\mathrm{do}(x)\\}$ is misspecified, the $\mathbb{P}\_{\mathrm{DC}}$ would not be accurate. As claimed in lines 283-284, 432-434 (right column), our method requires $p\\{y|\mathrm{do}(x)\\}$ to be identified correctly. A possible solution is to update with interventional data, such as setting $k\_0 \leq 1$. We will explore this in the future.
>
> **Difference between switch-light task and simulation.** The switch-light task different from the one in simlation, as it involves confounding bias caused by $S\_1 \rightarrow L\_1$, $S\_2 \rightarrow L\_2$ or $S\_1 \rightarrow L\_2$, $S\_2 \rightarrow L\_1$. We present this to illustrate the applicability of our mehtod to this setting.
>
> **About table 2.** As claimed in line 411, we only say our method is competitive to others. Overall, we achieve comparable and better perfomance than other baselines across all tasks.

---

### Official Review · Reviewer_yCr9 · 2025-03-12

**Overall Recommendation:** 3

**Summary:**

This paper presents a Bayesian active learning framework for identifying causal directions between variables through interventional strategies. Different from traditional information-theoretic approaches, it introduces an objective based on Bayes factors, which directly quantify the strength of evidence supporting one causal hypothesis over another. The authors formulate a sequential intervention objective that balances immediate and future evidential gains to optimize intervention selection.

To manage the computational complexity, they propose a dynamic programming algorithm that makes multi-step intervention planning tractable. The framework is tested across bivariate systems, tree-structured graphs, and embodied AI reasoning tasks, showing superior performance over baselines like InfoGain and random strategies. This approach effectively improves causal direction discovery, especially under limited budgets and weak causal influences.

[Attention: Kindly note that the paper slightly exceeds the 8-page main-paper limit by one line.]

**Claims And Evidence:**

Yes. The core part of this paper is the Bayesian active learning framework. Some preliminary information is clearly introduced in Section 3, and the core methodology is mentioned in Section 4 in details.

**Essential References Not Discussed:**

Yes. (1) The paper emphasizes decision-focused active learning via Bayes factors, but does not discuss classic Bayesian decision-theoretic foundations, for example, [Chick'2006]. (2) Quite some papers, about decision-making under causal uncertainty, are not mentioned in the paper, such as [Gonzalez'2018].

[Chick'2006] Chick, S.E. “Bayesian Approaches to Optimal Decision Making Under Uncertainty”, 2006.

[Gonzalez'2018] M. Gonzalez-Soto, et al. "Playing against Nature: causal discovery for decision making under uncertainty", 2018.

**Experimental Designs Or Analyses:**

Yes, the experiments mainly cover bivariate causal discovery and tree-structure causal graph learning as synthetic experiments, and also causal reasoning in embodied AI as real-world experiments.

**Methods And Evaluation Criteria:**

Yes, this paper is focusing on bivariate causal discovery task. The proposed method formulates this causal direction identification as a hypothesis-testing problem, and proposes a Bayes factor-based intervention strategy, which is interesting and totally making sense. As for the evaluation, it remains unclear why only Type I error and Recall (no considering Type II error, Precision, F1) are used in the evaluation.

**Other Comments Or Suggestions:**

Typos:

- Line 399: At Table 2, "Table 2. Average number of interventions required in the switch-ligh reasoning task" -> "switch-light reasoning task";

- Line 424: "We develope a simulation environment based on TongSim" -> "We develop a simulation ...";

**Other Strengths And Weaknesses:**

**Strengths:**

- With Bayes factor-based objective, this paper offers a direct and interpretable way to assess causal direction, addressing a major limitation of indirect mutual information-based methods.
- When sequential optimization meets with dynamic programming, it provides a theoretically grounded and computationally efficient way to plan multi-step interventions, outperforming greedy or myopic baselines.
- The proposed method is robust across settings. It demonstrates consistent performance in both weak and strong causation scenarios, making it suitable for diverse real-world applications.
- It is scalable to Multivariate Causal Discovery, extending naturally to tree-structured graphs and multivariate problems, and showing flexibility and generalization potential.

**Weaknesses and Comments:**
- Heavily relying on accurate prior estimation, this paper requires prior specification from observational data, and does not update priors using intervention data, which could limit adaptation to complex systems.
- Scalability to general graphs is still limited and challenging. While the method works well on tree structures, extension to general cyclic graphs or complex network structures remains a challenge.

**Questions For Authors:**

- Why only Type I error and Recall are used, while no considering Type II error, Precision, F1, in the experimental evaluations?

- How robust is the Bayes factor-based intervention strategy if the true data-generating process violates causal sufficiency?

- How sensitive is your method’s performance to these hyperparameters? e.g., $k_0$ = 10, $k_1$=0.1, how do you find those values?

**Relation To Broader Scientific Literature:**

Traditional approaches in active causal discovery have predominantly used information-theoretic objectives, while this paper shifts from information gain to Bayes factor-based objectives, which is somewhat a new aspect.

**Theoretical Claims:**

Yes, the only theoretical claims in this paper are Proposition 4.1 and Proposition 4.2. I have checked the correctness of these two Propositions.

---

> ### Author Rebuttal · Authors · 2025-04-01
>
> Thanks for your efforts and feedback in reviewing our paper. We address your questions below.
>
> **Why estimating priors using observation data?** We do not update priors using interventional data since if $k\_0 > 1$, it is difficult for $\mathrm{BF}\_{01}$ to be greater than $k\_0$ to identify $\mathbb{H}\_0$. To explain, even if $\mathbb{H}\_0$ is true, the data may exhibit a weak correlation between $X$ and $Y$. In such cases, it could happen that $\mathbb{P}(\mathcal{D}|\mathbb{H}\_1) \geq \mathbb{P}(\mathcal{D}|\mathbb{H}\_0)$, making it fail to identify $\mathbb{H}\_0$ with $k\_0 > 1$. A possible solution is to choose a lower $k\_0$, such as setting $k\_0 \leq 1$. We will explore this in the future.
>
> **Scalability to general graphs.** We can extend our method to the general graph by redefining $\mathbb{P}\_{\mathrm{DC}}$ as the ratio of maximum likelhood over the equivalence class:
>
> $$\mathrm{BF}\_{01} = \frac{\max\_{G \in \mathcal{G}^{(x,y)}\_0} \mathbb{P}\\{\mathcal{D}| G,\mathrm{do}(x),\mathbb{H}\_0\\}}{\max\_{G \in \mathcal{G}^{(x,y)}\_0} \mathbb{P}\\{\mathcal{D}| G,\mathrm{do}(x),\mathbb{H}\_1\\}}$$
>
> where $\mathcal{G}^{(x,y)}\_0$ (*resp.* $\mathcal{G}^{(x,y)}\_1$) denotes the Markov equivalence class regarding $p\\{y|\mathrm{do}(x)\\}$ under $\mathbb{H}\_0$ (*resp.* $\mathbb{H}\_1$). Since $\mathcal{G}^{(x,y)}\_0$ (*resp.* $\mathcal{G}^{(x,y)}\_1$) encompasses the true graph under $\mathbb{H}\_0$ (*resp.* $\mathbb{H}\_1$), it recovers the true likelihood ratio. We will pursue this direction in the future.
>
> **Evaluation metrics.** In our causal hypothesis testing setting, where $\mathbb{H}\_1$ (causal relationship exists) is the primary concern in causal discovery, we focus on Type I error rate (probability of detecting $\mathbb{H}\_1$ when $\mathbb{H}\_0$ is true) and recall (probability of detecting $\mathbb{H}\_1$ when $\mathbb{H}\_1$ is true, which is 1 - Type II error rate). We have provided all metrics (Type I error rate, Type II error rate, precision, recall, F1) at https://sites.google.com/view/additional-figures-icml4692/. Our method achieves competitive precision while outperforming baselines on other metrics. We will include these comprehensive results in the final version.
>
> **Robustness to causal sufficiency violations.** We assume causal sufficiency since our method requires the identifiability of $p\\{y|\mathrm{do}(x)\\}$, as claimed in lines 283-284, 432-434 (right column).
>
> To relax this condition, we can estimate $p\\{y|\mathrm{do}(x)\\}$ using interventional data. In this regard, we should adjust the threshold for $\mathrm{BF}\_{01}$, as discussed in our response to the first question.
>
> **Threshold of Bayes Factor.** We choose following [1,2] that provided evidence categories for different thresholds. Following [1,2], we chose $k\_0=10$ as the threshold for "strong evidence". We also conducted experiments with $k\_0=30, 100$ ("very strong" and "extreme evidence") available at https://sites.google.com/view/additional-figures-icml4692/ and the results verified the utility of our methods. We will include these experimental results in the final version.
>
> **Additional references.** Thanks for your suggestions, we will add these references and discuss them accordingly.
>
> **References**
>
> [1] Jeffreys, H. Theory of Probability (3rd Edition). Oxford, University Press, 1961.
>
> [2] Schönbrodt, F. D. and Wagenmakers, E.-J. Bayes factor design analysis: Planning for compelling evidence. Psychonomic bulletin & review, 25(1):128-142, 2018.

---

> > ### Comment · Reviewer_yCr9 · 2025-04-08
> >
> > Thanks for the detailed responses. Overall, I like the core idea of this paper, which formulates the causal direction identification as a hypothesis testing problem and makes use of Bayes factor-based intervention strategy. Therefore, I stand by my positive score.

---

### Official Review · Reviewer_g8jk · 2025-03-12

**Overall Recommendation:** 3

**Summary:**

This paper investigate the problem of determining the direction of relationships between variables by active intervention. The previous literature tries to maximize the information-theoretic gains for deciding the intervention value which may not effectively measure the reliability of direction determination. In contrast, this paper leverage the Bayes factor to formulate the objective function. To consider the multi-step intervention, this paper also propose a sequential intervention objective and develop a dynamic programming algorithm to solve it. The extensive experimental results show the effectiveness of the method in requiring less intervention and more correct direction.

**Claims And Evidence:**

I think the empirical comparison experiments and the method design support the claims. The method performs better in direction determination and can be applied in wide scenarios.

**Essential References Not Discussed:**

I think the main references have been discussed.

**Experimental Designs Or Analyses:**

I think the experimental design and analysis make sense and clearly shows the effectiveness of the method.

**Methods And Evaluation Criteria:**

I have some concerns about the design of methods. The details can be found in the Questions. The evaluation process makes sense.

**Other Comments Or Suggestions:**

No other comments.

**Other Strengths And Weaknesses:**

This paper investigate a solid and challenging problem. The proposed method is advanced, but left some concerns of the soundness. The details of the concern can be found in the questions.

**Questions For Authors:**

I have the following concerns about the method.

1. In equation (1b) and (1c), the probability is conditional on $D_m$. However, the definition of BF is not related to this condition. In contrast, in equation (6), this condition is missed.

2. In the next paragraph, how is the threshold of $\omega \frac{\gamma_0}{1-\gamma_0}$ derived?

3. In the objective function (8), the discount $\gamma$ is missed.

4. According to the Complexity Analysis subsection, the time complexity is exponential rather than polynomial.

**Relation To Broader Scientific Literature:**

This paper investigate a classical problem of determining the causal direction. The main contribution is concentrated on the novel objective functions developed with Bayes Factors and an improvement to multi-step version.

**Theoretical Claims:**

I check the proof. However, I think the statement in Proposition 4.2 have some flaws. "Given $\bar{x}_{-k}^*$"  should be modified as

"Given $\bar{x}_{1:k-1}^*$".

---

> ### Author Rebuttal · Authors · 2025-04-01
>
> We appreciate your efforts and positive feedbacks regarding our paper. We address your concerns below.
>
> **Definition of $\mathbf{x\_{-k}}$ in Proposition 4.2.** Thank you for your clarification. We will correct this in the updated version.
>
> **Conditional Probability in equations (1b), (1c), and (6).** As mentioned in lines 201-202 (right column), we omit $\mathcal{D}\_m$ for simplicity. We apologize for any confusion and will highlight this in the updated version.
>
> **Threshold derivation.** Thank you for pointing this out. The threshold should be $\frac{\gamma\_0}{(1-\gamma\_0)\omega}$. According to Bayes formula, we have:
>
> $$\mathrm{BF}\_{01} = \frac{\mathbb{P}(\mathcal{D}|\mathbb{H}\_0)}{\mathbb{P}(\mathcal{D}|\mathbb{H}\_1)} = \frac{\mathbb{P}(\mathbb{H}\_0|\mathcal{D})\mathbb{P}(\mathbb{H}\_1)}{\mathbb{P}(\mathbb{H}\_1|\mathcal{D})\mathbb{P}(\mathbb{H}\_0)}$$
>
> Since we require $\mathbb{P}(\mathbb{H}\_0|\mathcal{D}) > \gamma\_0$, and by $\omega:=\frac{\mathbb{P}(\mathbb{H}\_0)}{\mathbb{P}(\mathbb{H}\_1)}$, we should have $k\_0 = \frac{\gamma\_0}{(1-\gamma\_0)\omega}$.
>
> **Missing Discount Factor $\gamma$.** Thank you for pointing this out. We will correct it later.
>
> **Complexity Analysis.** The complexity is related to the number of intervention steps $K$, rather than $|\mathcal{X}|$. Since $X$ is discrete, the cardinality of the input space $|\mathcal{X}|$ is finite and remains constant once the setting is determined.

---

### Official Review · Reviewer_aqQB · 2025-03-14

**Overall Recommendation:** 3

**Summary:**

This paper focuses on causal discovery through Bayes factor. Instead of using information-theoretic gains to determine the direction of causal relationships, this paper adopts Bayes factor and formulating the task as a hypothesis testing. Furthermore, it uses sequential experiment design to selectively gather information for effective optimization.

## update after rebuttal: overall this work presents an interesting idea. The authors' responses help clarify my questions. I would like keep my positive score.

**Claims And Evidence:**

One claim is that: “we employ Bayes factors to establish an objective that directly optimizes the probability of obtaining decisive and accurate evidence, leading to more efficient causal discovery.”. This claim is supported by discussions on connection to information gain. However, it is not clear to me the real benefit by replacing traditional methods with Bayes factors.

**Essential References Not Discussed:**

N/A

**Experimental Designs Or Analyses:**

How does the proposed approach compare to (Castelletti & Consonni, 2024)?

**Methods And Evaluation Criteria:**

The proposed methods make sense to me. I think it is an interesting idea that formulating the task as a hypothesis testing problem.

**Other Comments Or Suggestions:**

N/A

**Other Strengths And Weaknesses:**

N/A

**Questions For Authors:**

1.	It is interesting to formulate the task as an independent test. Is there any justification in using Bayes Factor instead of other techniques for performing independence tests?

2.	How is the threshold of Bayes factor determined? Does the performance sensitive to the threshold?

3.	In Table 1, three different sizes of sample set are considered. Considering the bivariable case, these sizes should be sufficient. How about further reducing the size of samples? Will the model remain robust under insufficient data?

4.	For multivariate cases, the authors discussed 3-node graph in Appendix and studied tree-graph in the main paper. Are there any justifications like under which conditions the proposed approach work? In other words, how does the proposed approach generalize complex graphs with multiple variables and with loops.

5.	What is the complexity of the proposed approach compared to existing methods.

6.    How does the proposed approach compare to (Castelletti & Consonni, 2024) in practice?

**Relation To Broader Scientific Literature:**

Improving efficiency of causal discovery is important in many real-world applications, especially involving reasoning. This work could lay a strong foundation for empowering AI systems with causal reasoning.

**Theoretical Claims:**

I have checked the derivations showing connection to information gain.

---

> ### Author Rebuttal · Authors · 2025-04-01
>
> We appreciate your efforts and valuable feedback in reviewing our paper. Here are our responses.
>
> **About benefits of using Bayes Factor.** We choose the Bayes factor because it naturally aligns well with our goal and hence is more efficient for optimization. Bayes factors are commonly used in experimental design to assess the strength of competing hypotheses. When expressed in terms of $\mathbb{P}\_{\mathrm{DC}}$, it directly supports our objective of maximizing certainty and correctness in causal discovery. While maximizing $\mathbb{P}\_{\mathrm{DC}}$ can also increase the information gain, it is a transformation of $\mathbb{P}\_{\mathrm{DC}}$, and optimizing this transformation is less efficient. As shown in Figures 1 and 3, $\mathbb{P}\_{\mathrm{DC}}$ offers higher intervention efficiency compared to information gain.
>
> **Threshold of Bayes Factor.** We choose following [1,2] that provided evidence categories for different thresholds. Following [1,2], we chose $k\_0=10$ as the threshold for "strong evidence". We also conducted experiments with $k\_0=30, 100$ ("very strong" and "extreme evidence") available at https://sites.google.com/view/additional-figures-icml4692/ and the results verified the utility of our methods. We will include these experimental results in the final version.
>
> **Performance with small sample sizes.** We tested our method with $n=100$ and found it remains robust compared to the result with $n=1,000$. Below are the results for the Type I error rate (the proportion of cases where $\mathrm{BF}\_{10} > k\_1$ under $\mathbb{H}\_0$, which differs from the more conservative definition used in the paper: $\mathrm{BF}\_{10} > \frac{1}{k\_0}$ under $\mathbb{H}\_0$) and recall (the proportion of cases where $\mathrm{BF}\_{10} > k\_1$ under $\mathbb{H}\_1$):
>
> |$\|\mathcal{D}\_{\mathrm{obs}}\|$|Type I Error (Setting 1)| Recall (Setting 1) | Type I Error (Setting 2)|Recall (Setting 2)|
> |----------------------| ------------------------|------------------|------------------------|------------------|
> |$100$| $0.0205\pm0.0128$|$0.7595\pm 0.0483$|$0.0005\pm 0.0022$|$1.0000\pm 0.0000$|
> |$1000$| $0.0305\pm0.0186$|$0.8840\pm 0.0325$|$0.0000\pm 0.0000$|$1.0000\pm 0.0000$|
>
> These results show our method maintains high performance even with significantly reduced observational data.
>
> **Applicability to more complex graphs.** As discussed in lines 283-284, our current method requires $p\\{y|\mathrm{do}(x)\\}$ to be identifiable from observational data, in order to calculate $\mathbb{P}\_{\mathrm{DC}}$. As claimed in the "Discussion" section, this might not be achievable in multivariate causal discovery, as $p\\{y|\mathrm{do}(x)\\}$ cannot be uniquely determined among the Markov equivalent class.
>
> To resolve this problem, a promising solution is to extend $\mathbb{P}\_{\mathrm{DC}}$ as the ratio of maximum likelhood over the equivalence class:
>
> $$\mathrm{BF}\_{01} = \frac{\max\_{G \in \mathcal{G}^{(x,y)}\_0} \mathbb{P}\\{\mathcal{D}| G,\mathrm{do}(x),\mathbb{H}\_0\\}}{\max\_{G \in \mathcal{G}^{(x,y)}\_0} \mathbb{P}\\{\mathcal{D}| G,\mathrm{do}(x),\mathbb{H}\_1\\}}$$
>
> where $\mathcal{G}^{(x,y)}\_0$ (*resp.* $\mathcal{G}^{(x,y)}\_1$) denotes the Markov equivalence class regarding $p\\{y|\mathrm{do}(x)\\}$ under $\mathbb{H}\_0$ (*resp.* $\mathbb{H}\_1$). Since $\mathcal{G}^{(x,y)}\_0$ (*resp.* $\mathcal{G}^{(x,y)}\_1$) encompasses the true graph under $\mathbb{H}\_0$ (*resp.* $\mathbb{H}\_1$), the maximum value recovers the true likelihood. We will pursue this direction in the future.
>
> **Computational Complexity.** Existing methods [3] adopted the greedy strategy. To avoid the local minima issue[4], they need to optimize the multi-step objective. The complexity is exponential. By using dynamic programming, we can reduce it to polynomial complexity.
>
> **Comparison with Castelletti & Consonni (2024).** The comparison may not be appropriate as we consider different settings. Castelletti & Consonni (2024) focuses on pre-experiment sample size determination without interaction with the environment. In contrast, we focus on determining the intervention value and can receive feedback from the environment at each iteration.
> **References**
>
> [1] Jeffreys, H. Theory of Probability (3rd Edition). Oxford, University Press, 1961.
>
> [2] Schönbrodt, F. D. and Wagenmakers, E.-J. Bayes factor design analysis: Planning for compelling evidence. Psychonomic bulletin & review, 25(1):128-142, 2018.
>
> [3] Toth C, Lorch L, Knoll C, et al. Active bayesian causal inference[J]. Advances in Neural Information Processing Systems, 2022, 35: 16261-16275.
>
> [4] Greenewald, K., Katz, D., Shanmugam, K., Magliacane, S., Kocaoglu, M., Boix Adsera, E., and Bresler, G. Sample efficient active learning of causal trees. Advances in Neural Information Processing Systems, 32, 2019.

---

### Decision · Program_Chairs · 2025-05-01

**Decision:**

Accept (poster)

**Comment:**

The reviewers agree on this paper, in details they found strong points in the manuscript; i) the paper investigates a solid and challenging problem, and the proposed method is advanced, ii) the idea to use the Bayes factor-based objective, offers a direct and interpretable way to assess causal direction, addressing a major limitation of indirect mutual information-based methods, iii) the combination of sequential optimization and dynamic programming, provides a theoretically grounded and computationally efficient way to plan multi-step interventions, outperforming greedy or myopic baselines, iv) the performance under several settings is robust as it emerges from the numerical experiments, v) the proposed method is scalable to multivariate causal discovery, thus widening the settings where the method can be fruitfully applied. However, some weacknesses were pointed out, the main being; i) some criticism exists on the the soundness, ii) the paper strongly  relies on accurate prior estimation, while the proposed method requires prior specification from observational data, and does not update priors using intervention data, and this has been pointed out to limit adaptation to complex systems, iii) Some limitations was also raised with respect to scalability to general graphs. Indeed, while the method works well on tree structures, extension to general cyclic graphs or complex network structures remains a challenge. iv) anothe limitation was that the experimental setting seems a bit weak, in that all the experiments provided essentially only focus on the case of binary variables and Bernoulli distribution. However, the rebuttals from the author, as recognized by almost all the reviewers, solved all the concerns that were in the reviews. I personally liked the authors rebuttals which addressed point by point all the critical aspects as spotted by the reviewers.